# A LARGE SCALE ANALYSIS OF GENDER BIASES IN TEXT-TO-IMAGE GENERATIVE MODELS

## ABSTRACT

With the increasing use of image generation technology, understanding its social biases, including gender bias, is essential. This paper presents a large-scale study on gender bias in text-to-image (T2I) models, focusing on everyday situations. While previous research has examined biases in occupations, we extend this analysis to gender associations in daily activities, objects, and contexts. We create a dataset of 3,217 gender-neutral prompts and generate 200 images over 5 prompt variations per prompt from five leading T2I models. We automatically detect the perceived gender of people in the generated images and filter out images with no person or multiple people of different genders, leaving 2,293,295 images. To enable a broad analysis of gender bias in T2I models, we group prompts into semantically similar concepts and calculate the proportion of male- and female-gendered images for each prompt. Our analysis shows that T2I models reinforce traditional gender roles and reflect common gender stereotypes in household roles. Women are predominantly portrayed in care and human-centered scenarios, and men in technical or physical labor scenarios. Code and data will be released.

## 1 INTRODUCTION

Rapid advances in image generation technology make it easier than ever to automatically generate large amounts of synthetic images. State-of-the-art text-to-image (T2I) models (Black Forest Labs, 2024; Stability AI, 2024) can generate high-quality images from arbitrary text instructions. Their capabilities are further enhanced through editing (Brooks et al., 2023; Kawar et al., 2023; Zhang et al., 2024b) and personalization (Ruiz et al., 2023; Gal et al., 2023; Kim et al., 2024; Bini et al., 2024) techniques. Synthetic images are not only used in everyday applications such as advertisements (Lin et al., 2023) and presentation slides (Peng et al., 2024) but are also increasingly used as training data for other foundation models (Tian et al., 2024a;b; Fan et al., 2024; Kim et al., 2024). Here, Chen et al. (2024a) have observed age and skin tone bias amplification at increased levels of synthetic images in the pretraining data.

As the availability and proliferation of synthetic images increase, so does their power to influence society and amplify any harms originating from the underlying models (Chan et al., 2023). In their seminal work, Buolamwini & Gebru (2018) discovered intersectional gender and racial biases in image recognition systems. Within the research community, the list of known biases has only grown: Agarwal et al. (2021); Hall et al. (2024); Berg et al. (2022); Seth et al. (2023); Tanjim et al. (2024) identified social biases in CLIP (Radford et al., 2021), such as associating men with words related to criminal activities. Hendricks et al. (2018); Hirota et al. (2022; 2023) found social biases in automatic image captioning. Zhang et al. (2024a); Xiao et al. (2024); Girrbach et al. (2025); Fraser & Kiritchenko (2024); Ruggeri et al. (2023) uncovered various social biases in multimodal large language models (MLLMs). These examples demonstrate that social bias pervades all aspects of modern generative AI systems.

Research on social bias in T2I models has led to a large body of work covering all computational aspects of social bias, including bias analysis (Cho et al., 2023; Bianchi et al., 2023; Luccioni et al., 2024), open bias detection (D'Incà et al., 2024; Chinchure et al., 2024; Dehdashtian et al., 2025), and model debiasing (Zhang et al., 2023; Bansal et al., 2022; Esposito et al., 2023; Friedrich et al., 2024). However, most research in this area has focused on gender-occupation bias (Wan et al., 2024). While this is an important issue, other aspects of daily life, such as everyday activities and stereotypical

contexts, also contribute to perpetuating or amplifying harmful social biases and require careful analysis. Furthermore, many studies on social bias in T2I models use a limited set of prompts and only a few images per prompt, reducing their representativeness.

We present a large-scale, in-depth study on gender bias in T2I models concerning everyday scenrios and address gaps in the literature by analyzing everyday activities complemented by gender-object and gender-context associations. Our main contributions are: (1) We compile 3,217 gender-neutral prompts over four categories to probe T2I models for gender bias and generate 200 images from five state-of-the-art T2I models for each prompt and filter unsuitable images, leaving 2,293,295 images for analysis; (2) We design a carefully structured experimental setup to analyze gender bias in large image datasets and systematically examine the observed gender biases and relate them to known human gender biases; (3) We analyze bias amplification in activities compared to LAION-400m and confirm bias amplification in occupations wrt. U.S. labor statistics. Our work takes an important step toward addressing gender stereotypes in T2I models, helping understand and document perpetuation of gender inequality in AI technology (Tannenbaum et al., 2019; Abebe et al., 2020).

## 2 RELATED WORK

Work analyzing social bias in T2I generation has largely targeted a few categories (Wan et al., 2024), especially perceived gender and race, and mostly used occupation prompts. In their seminal work, Bianchi et al. (2023) show bias amplification in occupations and personal attributes but examine only 20 manually curated prompts. We provide a larger-scale, automated analysis offering broader, more precise insights. Cheong et al. (2024) study gender and racial bias in occupations and report stronger gender imbalance than U.S. labor statistics; we confirm and extend this by analyzing activities as well. Luccioni et al. (2024) measure bias without explicit gender/race labels, identifying distributional differences across 4,380 images from 146 occupation prompts, and explicitly call for deeper studies, which we conduct in this work. Similarly, Cho et al. (2023) observe differences in 737 images from 83 occupation prompts using gendered vs. gender-neutral phrasing. In contrast, we sample more images, filter and crop unsuitable ones, and consider scenarios beyond occupations.

Previously, Zhang et al. (2024c); Wu et al. (2024); Mannering (2023) investigate gender bias in person–object co-occurrence, but on narrow object sets (mainly clothing). We demonstrate gender bias in contexts such as traditional household roles with substantial societal relevance. Ungless et al. (2023) show poor representation of non-binary identities; we exclude them due to conceptual and technical limitations (Appendix J). Likewise, Ghosh & Caliskan (2023) show weak representation of national identities and overly sexualized depictions of women, especially for Global South prompts; Jha et al. (2024) quantify national stereotyping. Wu et al. (2024) generate 800,000 images from 200,000 gender-neutral prompts, but few images per prompt limit conclusions about which specific biases arise and their magnitude.

Although previous work establishes gender bias, scenarios beyond occupations remain underrepresented (Wan et al., 2024). Most studies are small-scale, typically under 200 prompts with no more than 20 images per prompt. Even larger benchmarks (Luo et al., 2024; 2025) still center on occupation prompts. Accordingly, we answer calls for in-depth analyses across a broader range of scenarios (Wan et al., 2024; Luccioni et al., 2024) and document the models' default "worldview" (De Simone et al., 2023; Katirai et al., 2024). Because many template-based prompts yield images without clearly gendered people (Lyu et al., 2025), we ensure validity by filtering images without people or depicting both men and women (see Appendix K).

## 3 PROMPTS AND IMAGES

### 3.1 PROMPT COLLECTION, PROCESSING AND CLUSTERING

We collect prompts in four categories: (1) Activities, (2) Contexts, (3) Objects, and (4) Occupations. To study gender bias in everyday activities, we rely on the curated set from (Wilson & Mihalcea, 2017), i.e. **Activities**. The activities in (Wilson & Mihalcea, 2017) were gathered through Amazon Mechanical Turk from U.S. based workers, who were asked to provide short phrases describing recent activities. Including these 1405 activities provides insight into how stereotypical gender roles in everyday life are reflected in T2I models. Analyzing gender associations with contexts and objects

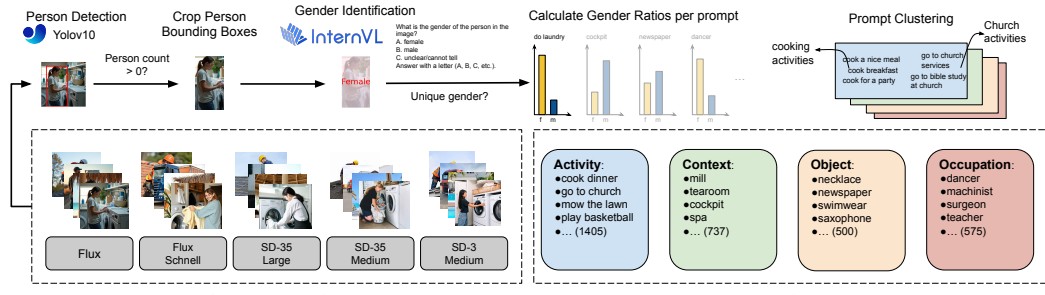

Figure 1: Overview of our experimental setup to analyze gender bias in T2I models regarding everyday scenarios. We show our 4 prompt groups on the bottom right and the five T2I models on the bottom left. The top left visualizes our filtering method: First, we detect people, crop the bounding boxes, and detect perceived gender. We remove images without people or showing at least one man and one woman. We calculate the proportions of female- and male-labeled images generated for each prompt and analyze systematic gender biases.

| Activity | "a person who is {{verb}}-ing {{activity}}" | Context | "a person {{in/on/at...}} the {{context}}" |
|---|---|---|---|
| Objects | "a person and {{a object}}" | Occupations | "a person working as {{occupation}}" |

Table 1: Prompt templates for the different prompt groups. Parts in double brackets {{...}} are modified or filled in by the LLM.

further extends this analysis by considering everyday situations beyond specific activities. To study gender bias in relation to people and places, i.e. **Contexts**, we collect a set of 737 scene classes from the SUN Database (Xiao et al., 2010; 2016). We include all classes but do not consider fine-grained distinctions, e.g. "inside" the church and "outside" the church leads to the context "church".

We collect 500 common physical objects from WordNet Fellbaum (1998), i.e. **Objects**. To select these 500 objects, we filter all noun hyponyms of the `object.n.01` WordNet synset using a list of the most common English words. Additionally, we manually remove a small number of unsuitable synsets, e.g. synsets that refer to people or body parts, or places that are already covered in the contexts prompt group. We retain the top 1000 most frequent lemmas and re-rank them based on their concreteness following Brysbaert et al. (2014). Finally, we select the top 500 most concrete lemmas from the top 1000 most frequent WordNet lemmas. We also include occupations to align with prior work and to examine occupation-related gender bias in T2I models. Our occupation list is comparatively large, as we include all 575 occupations listed by Bureau of Labour Statistics (2023) rather than a subset, i.e. **Occupations**.

Using `Yi-1.5-34B` (Young et al., 2024) (comparison in Appendix C.2), we convert collected activities, contexts, objects, and occupations into syntactically coherent prompts. Prompts are gender-neutral and begin with "a person". We apply group-specific templates (Table 1); the LLM prompts used to fill them are in Appendix C.1. We also simplify occupation descriptions, as (Bureau of Labour Statistics, 2023) is often overly detailed. We generate 5 variations per prompt by replacing the prefix "a person" with "an individual", "someone", "a friend", and "a colleague", which do not include any gender information. Prompt variations increase diversity and are essential for valid analysis of T2I social bias (Seshadri et al., 2022; Sclar et al., 2024; Hida et al., 2024). Appendix D.4 shows variations do not lead to significant skew towards male- or female-gendered images.

For a concise presentation of our findings from the 3217 prompts, we cluster prompts in all four prompt groups into semantically coherent concepts. We apply the following variation of BERTopic (Grootendorst, 2022) to cluster prompts. First, we embed all prompts using a sentence embedding model (Reimers & Gurevych, 2019). Then, we reduce the embedding dimensions to 16 using UMAP (McInnes et al., 2018). Using HDBSCAN clustering (McInnes & Healy, 2017) with cosine distance, we determine the concept clusters, adding unclustered prompts by HDBSCAN to the cluster with

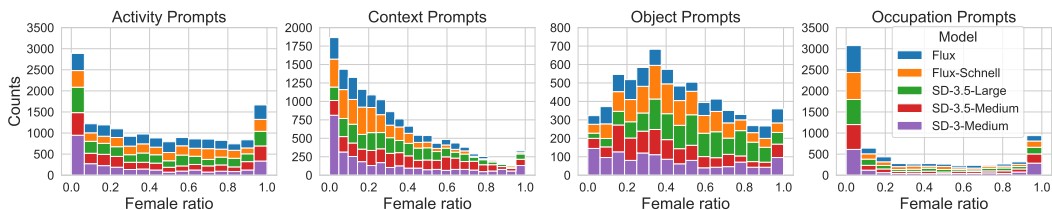

Figure 2: Stacked distribution of female ratios in generated images for all models and prompt groups.

the nearest centroid. The detailed settings are in Appendix C.3. We arrive at 165 activity clusters, 91 context clusters, 62 object clusters, and 76 occupation clusters.

After clustering prompts, we summarize all clusters by an LLM (see Appendix C.3). Summarizing a list of prompts requires more in-depth understanding and reasoning than prompt processing, so we use `Llama-3-3-70B-Instruct` based on manual inspection. A comparison to other LLMs and our exact prompt template is in Appendix C.3. For the purposes of our analysis, we further merge clusters with the same summary (e.g. different variants of shopping-related activities), as they would be indistinguishable for the reader. However, our code release will include the fine-grained clusters.

## 3.2 IMAGE GENERATION AND GENDER IDENTIFICATION

We generate images using 5 models, which represent the state-of-the-art among open models at the time of writing: (1) Flux (Black Forest Labs, 2024), (2) Flux-Schnell, (3) Stable Diffusion 3.5 Large (Stability AI, 2024), (4) Stable Diffusion 3.5 Medium, and (5) Stable Diffusion 3 Medium (Esser et al., 2024). These are latent diffusion models (Rombach et al., 2022) based on Diffusion Transformers (Peebles & Xie, 2023). Other recent strong models, such as Flux-Krea and Qwen-Image (Wu et al., 2025), were concurrently released to this study and unavailable when conducting experiments. For each combination of model and prompt, we generate 40 images per prompt variation. This results in 5 models $\times$ 5 prompt variations $\times$ 3217 prompts $\times$ 40 images = 3,217,000 images.

To analyze gender bias in generated images, we identify the perceived gender of the people shown in the images using a two-step process. First, we detect all people in the images using the object detector `YOLOv10` (Wang et al., 2024) and obtain bounding boxes for the detected individuals. Next, we crop each detected person's bounding box and pass it to an MLLM, `InternVL2-8B` (Chen et al., 2024b;c), along with a prompt asking the model to identify the person's gender as "female", "male", or "unclear/cannot tell". We focus on binary perceived gender, specifically men and women, for several reasons. First, it is unclear whether non-binary gender has distinct visual representation, in any case current T2I models do not generate features that clearly indicate non-binary gender. Second, current MLLMs do not consider non-binary gender as an option, as shown in Appendix E.2. While not addressed in this paper, the fact that models output gender as binary is a separate issue that warrants further discussion.

Using bounding boxes instead of the full image helps mitigate bias from the person's context, i.e. predicting the gender based on the background and not the person's features, as MLLMs also exhibit gender bias (Girrbach et al., 2025). It also prevents confusion when multiple people of potentially different genders appear in the image. In Appendix E.1, we provide the detailed prompt used for gender identification and verify, using human-labeled data, that the MLLM used in this study can identify perceived gender with near-perfect accuracy. It is important to note that assigning a person's gender can be problematic, because it cannot necessarily be perceived from an image, and gender is a spectrum. Therefore, good practice with images of real people is to have people self-identify their gender. However, T2I models create images that are not real, so assigning perceived gender to the images is more acceptable as there is no risk of misidentifying a real person.

## 3.3 IMAGE AND PROMPT FILTERING

We exclude images and prompts that are not suitable for analyzing gender bias, i.e. if (a) There is no person in the image; (b) There is no person in the image whose gender can be clearly identified (see Appendix E.3 for details); (c) There are multiple people in the image and there is at least one man

| # Prompts (# Imgs.) | Activities | | Contexts | | Objects | | Occupations | |
|---|---|---|---|---|---|---|---|---|
| | 1405 | (1,405K) | 737 | (737K) | 500 | (500K) | 575 | (575K) |
| Flux | 1149 | (187,079) | 632 | (100,125) | 395 | (58,387) | 574 | (110,989) |
| Flux-Schnell | 1096 | (168,994) | 693 | (105,737) | 466 | (63,520) | 574 | (105,288) |
| SD-3.5-large | 1163 | (185,310) | 689 | (113,839) | 476 | (77,789) | 570 | (108,454) |
| SD-3.5-Medium | 1076 | (163,845) | 693 | (112,296) | 411 | (59,507) | 572 | (107,691) |
| SD-3-Medium | 1062 | (169,403) | 711 | (124,520) | 418 | (62,702) | 573 | (107,820) |

Table 2: Prompt groups with remaining prompts and images in brackets after filtering.

and one woman. If there are multiple people, we keep the image if people with all the same gender are shown or where the gender of other people is labeled "unclear/cannot tell." (for example people in the background). Additionally, we exclude entire prompts for a model if fewer than 100 out of 200 images remain after filtering. Since we analyze gender bias at the prompt level, we can only consider prompts where we can reliably estimate the ratio of male and female people in the images. The number of remaining prompts for each model is in Table 2.

## 4 GENDER BIAS ANALYSIS EXPERIMENTS

For each prompt, e.g. the Activities prompt group has 1405 prompts, we calculate the ratio of male and female images. Let $\mathcal{I}^p = \{I_1^p, I_2^p, \ldots, I_n^p\}$, $n \leq 200$ be the set of gendered images for prompt $p$ after filtering and $\mathcal{G} : \mathcal{I} \to \{\text{female}, \text{male}\}$ the mapping of images to unique genders according to `InternVL2-8B`. Then, we define the female- and male-gendered images $F(p)$ and $M(p)$ as

$$F(p) := \{I \in \mathcal{I}^p \mid \mathcal{G}(I) = \text{female}\} \tag{1}$$

$$M(p) := \{I \in \mathcal{I}^p \mid \mathcal{G}(I) = \text{male}\} \tag{2}$$

and the female ratio $\mathcal{R}_f(p)$ of prompt $p$ as

$$\mathcal{R}_f(p) := \frac{|F(p)|}{|F(p)| + |M(p)|}. \tag{3}$$

This allows us to estimate the distribution of female ratios across activities for a given model, i.e. we present the distribution of values of $\mathcal{R}_f$ as a histogram in Fig. 2. Overall, we find that the models generate similar gender ratios across all prompts (see Appendix F.1 for more details). However, we also observe that models tend to generate more male-gendered images, as also observed in (Ghosh & Caliskan, 2023; Zhang et al., 2023).

### 4.1 ACTIVITIES AND CONTEXTS

In Fig. 2, we find a large number of activities in which the set of images is male-dominated with $\mathcal{R}_f$ close to zero. We say a prompt or a prompt cluster is female-dominated (male-dominated) if $\mathcal{R}_f \geq 0.7$ ($\mathcal{R}_f \leq 0.3$), i.e. 70% (30%) or more (less) of images for this prompt or cluster are female-gendered. If $\mathcal{R}_f$ is between 50% and 70%, we speak of female-leaning clusters or prompts (equivalently male-leaning). While gender ratios of activities are distributed more evenly, the distribution of gender ratios for contexts is heavily skewed toward male images. This highlights a general trend to outputting males overall and men as the default. We calculate the top 10 (top 5) activities with the highest ratio of female- or male-gendered images across T2I models to showcase male- and female-dominated activities (contexts). We state the summary and the per-model average ratio of female- or male-gendered images of activities (contexts) in the cluster for each activity (context) cluster. Results are in Fig. 3. For each cluster, we also report the mean $\mathcal{R}_f$ across models ($\hat{\mathcal{R}}_f$).

***Most female-dominated activities.*** Common female-dominated activities are *crafting* ($\hat{\mathcal{R}}_f \approx 85\%$), which comprises activities such as "crocheting" or "making bracelets", and pet-related activities (*cat* ($\hat{\mathcal{R}}_f \approx 79\%$), *pet* ($\hat{\mathcal{R}}_f \approx 81\%$)). *birthday* ($\hat{\mathcal{R}}_f \approx 83\%$) contains activities related to parties. Further care-related activities are *baby care* ($\hat{\mathcal{R}}_f \approx 88\%$) and *volunteer* ($\hat{\mathcal{R}}_f \approx 82\%$), which are both ex-amples of helping other people. The prominence on care-related activities reflects gender norms of

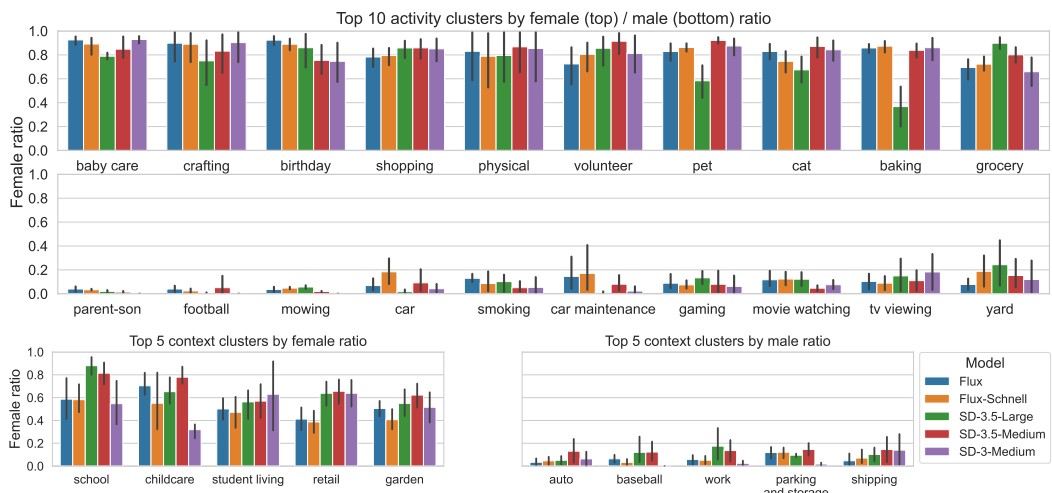

Figure 3: (Top) Top 10 most female-dominated (top row) and top 10 most male-dominated (bottom row) activity clusters. Bars indicate ratio of female-gendered images generated from 5 T2I models averaged over prompts in each cluster. Error line indicates the std. dev. across prompts. (Bottom) Top 5 most female-dominated (left) and top 5 most male-dominated (right) context clusters.

women as caretakers and resembles human stereotypes, as women are described as "warm", "sensitive to others", and specifically "interested in children" (Rudman et al., 2012). The remaining highly female-dominated clusters in Fig. 3 are shopping-related activities (*shopping* ($\hat{\mathcal{R}}_f \approx 83\%$), *grocery* ($\hat{\mathcal{R}}_f \approx 76\%$)) and yoga-related activities in *physical* ($\hat{\mathcal{R}}_f \approx 83\%$). Shopping-related activities include both shopping for daily necessities and clothes. Grocery shopping is known to be a household activity typically performed by women (Coltrane, 1989), and clothes are also associated with women in other prompt groups, especially contexts and objects (Section 4.2). Yoga was found to be seen as strongly female-typed (Matteo, 1986). Finally, "baking" ($\hat{\mathcal{R}}_f \approx 76\%$) is a female-typed way of cooking (Rokicki et al., 2016).

***Most male-dominated activities.*** One common male-dominated activity type in Fig. 3 is outdoor household work (*mowing* ($\hat{\mathcal{R}}_f \approx 3\%$), *yard work* ($\hat{\mathcal{R}}_f \approx 16\%$)), which includes "mowing the lawn", "cutting wood", and "raking leaves". Mowing the lawn specifically was identified as an activity typically performed by men (Coltrane, 1989). Further male-dominated activities are car-related (*car*, *car maintenance* (both $\hat{\mathcal{R}}_f \approx 8\%$)), which is also male-typed (Coltrane, 1989), as well as media consumption, such as (computer) *gaming* ($\hat{\mathcal{R}}_f \approx 9\%$) and *movie watching* ($\hat{\mathcal{R}}_f \approx 10\%$) or *tv viewing* ($\hat{\mathcal{R}}_f \approx 13\%$). According to Hilbrecht et al. (2008), young men, on average, devote more time to "watching TV and video" and "computer games" than women of the same age. However, Shaw (2012); Paaßen et al. (2017) find that (computer) gamers being predominantly male is more of a stereotype than reality, meaning that T2I models perpetuate the marginalization of women in e-sports (Paaßen et al., 2017). *Smoking* ($\hat{\mathcal{R}}_f \approx 8\%$) explicitly refers to cannabis consumption, which, alongside other drug consumption including alcohol, is more common among men than women (Wilsnack et al., 2000; Schulte et al., 2009; Hemsing & Greaves, 2020). *Football* ($\hat{\mathcal{R}}_f \approx 2\%$) is a strongly male-typed sport (Plaza et al., 2017) and also strongly male-dominated in T2I models. Many male-gendered images in the *parent-son* ($\hat{\mathcal{R}}_f \approx 2\%$) cluster are less surprising as prompts contain gendered words, i.e. "son".

***Most female-dominated contexts.*** The only consistently female-leaning clusters are *school* ($\hat{\mathcal{R}}_f \approx 68\%$) and *student living* ($\hat{\mathcal{R}}_f \approx 55\%$), describing university environments, such as "classroom", as well as *childcare* ($\hat{\mathcal{R}}_f \approx 60\%$), containing places such as "playroom". Teachers is a profession with female majority (in the USA, see (Bureau of Labour Statistics, 2023)), which could explain the association of women with school places. *Retail* ($\hat{\mathcal{R}}_f \approx 55\%$) contains various shopping locations, of which only a subset is strongly female-dominated. Such locations are related to fashion, such as "jewelry shop" or "hat shop". Details on the *retail* cluster are in Appendix H.1.

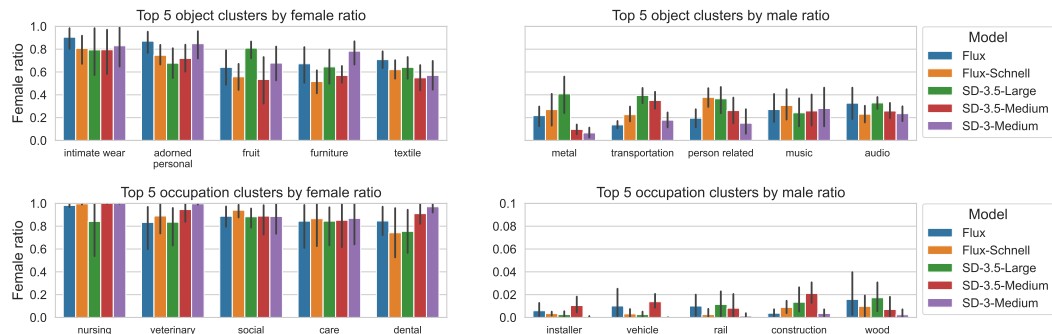

Figure 4: (Top) Top 5 most female-dominated (left) and top 5 most male-dominated (right) object clusters. (Bottom) Top 5 most female-dominated (left) and top 5 most male-dominated (right) occupation clusters (note that here the y-axis is between 0% and 10%).

***Most male-dominated contexts.*** In contrast to female-dominated contexts, the most male-dominated clusters focus on transportation (*auto* ($\hat{\mathcal{R}}_f \approx 7\%$), *shipping* ($\hat{\mathcal{R}}_f \approx 10\%$), *parking and storage* ($\hat{\mathcal{R}}_f \approx 10\%$)) and industrial places (*work* ($\hat{\mathcal{R}}_f \approx 9\%$)). Another strongly male-dominated cluster is *baseball* ($\hat{\mathcal{R}}_f \approx 7\%$), which contains "baseball field" and various locations therein, such as "batting cage" or "pitcher's mound". We note that baseball is a male-leaning sport (Matteo, 1986). It is clear that T2I models do not associate women with industrial, work-related places, and places where women are depicted seem to be social places, such as schools or certain shops. In Appendix I.3, we provide further analyses of workplace gender bias.

## 4.2 OBJECTS AND OCCUPATIONS

In Fig. 2, we observe a roughly Gaussian distribution for objects, peaking at a 0.4 female ratio. There are relatively few objects where models generate exclusively male- or female-gendered images. Gender distributions for occupations are highly polarized, with most occupations yielding only male-gendered images. Compared to actual work participation statistics, this reflects a clear bias amplification, in line with the findings by Seshadri et al. (2024). However, it also means that most previous work focusing on occupations has studied a particularly extreme example of bias amplification in T2I models. This further justifies our focus on activities and other everyday contexts. We list the top 5 object (occupation) clusters by female- and male ratios for each T2I model in Fig. 4.

***Most female-dominated objects.*** Main theme in female-dominated objects is clothing and accessories. *Adorned personal* ($\hat{\mathcal{R}}_f \approx 77\%$) contains different types of jewelry, such as "crystal" or "ring". *Furniture* ($\hat{\mathcal{R}}_f \approx 64\%$) and *textile* ($\hat{\mathcal{R}}_f \approx 62\%$) also fit this category, containing soft and textile-related objects such as "pillow" or "silk". "Intimate wear" ($\hat{\mathcal{R}}_f \approx 83\%$) contains underwear and swimwear. Another female-leaning cluster, particularly in SD models, is "fruit" ($\hat{\mathcal{R}}_f \approx 63\%$).

***Most male-biased objects.*** Common male-dominated objects are audio speakers (*audio*, $\hat{\mathcal{R}}_f \approx 28\%$) and music instruments (*music*, $\hat{\mathcal{R}}_f \approx 27\%$), vehicles (*transportation*, $\hat{\mathcal{R}}_f \approx 26\%$), and metal objects (*metal*, $\hat{\mathcal{R}}_f \approx 21\%$). We see a clear contrast between female-dominated objects, which are fashion-related, and male-dominated objects, which are technical. The male dominance in musical instruments is unexpected, as they oppose existing gendered associations of certain musical instruments (Abeles & Porter, 1978; Abeles, 2009). While we see these gender associations reflected in higher female ratios relative to other instruments (see Appendix H.2), musical instruments remain male-dominated.

***Most female-dominated occupations.*** We find many relations to previously discussed female-dominated prompt clusters. For example, *veterinary jobs* ($\hat{\mathcal{R}}_f \approx 90\%$) echo the pet-care-related activities, which we show are strongly female-dominated in Section 4.1. *Nursing jobs* ($\hat{\mathcal{R}}_f \approx 96\%$), *care jobs* ($\hat{\mathcal{R}}_f \approx 86\%$), and *social jobs* ($\hat{\mathcal{R}}_f \approx 90\%$) all describe human-centered caring activities. *Dental jobs* ($\hat{\mathcal{R}}_f \approx 84\%$) contains 4 occupations: "dental hygienist" and "dental assistant"

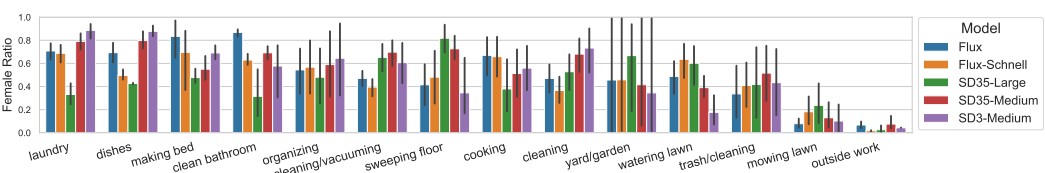

Figure 5: Household-related clusters of activity prompts.

are strongly female-dominated across all T2I models. "Dental technician" and "dentist" are female-dominated for all models except Flux-Schnell and SD-3.5-Large (see Appendix H.3 for detailed values). In the U.S., "dental hygienist" and "dental assistant" are occupations where $> 90\%$ of the workforce are women, whereas $\approx 59\%$ of dental technicians and $\approx 40\%$ of dentists are women.

***Most male-dominated occupations.*** While many occupations are male-dominated, the most male-dominated occupations are blue-collar jobs involving physical labor. This is, for example, the case with *installer jobs* ($\hat{\mathcal{R}}_f \approx 0\%$), *construction jobs* ($\hat{\mathcal{R}}_f \approx 1\%$), and *wood jobs* ($\hat{\mathcal{R}}_f \approx 1\%$). *Wood jobs* comprises occupations such as "carpenter" and "sawing machine operator". Similarly, *vehicle jobs* ($\hat{\mathcal{R}}_f \approx 1\%$) and *rail jobs* ($\hat{\mathcal{R}}_f \approx 1\%$) are transportation industry occupations. In contrast to female-dominated occupations, none of the top male-dominated occupations are human-centric.

## 4.3 SPECIAL TOPICS

We now look closely at specific topics of the activity prompt group, which have particular societal impact, namely household activities and bias amplification. More analyses on work/money-related activities are in Appendix I.1 and on bias amplification in jobs in Appendix I.5.

***Household Chores.*** The division of household chores between spouses in heterosexual marriages is strongly moderated by gender (Coltrane, 1989; Hiller & Philliber, 1986; Cerrato & Cifre, 2018; Kroska, 2003) and is relatively constant over time (Douthitt, 1989). To select a subset of household-chores-related clusters, we classify all prompts in the activities prompt group as representing a household chore or not by an LLM (see Appendix I.2), specifically `Phi-4`. We cluster the resulting 105 activities and get 14 clusters, which we label manually and plot in Fig. 5.

The seven clearly female-leaning clusters are *laundry* ($\hat{\mathcal{R}}_f \approx 68\%$), *dishes* ($\hat{\mathcal{R}}_f \approx 66\%$), *making bed* ($\hat{\mathcal{R}}_f \approx 65\%$), *clean bathroom* ($\hat{\mathcal{R}}_f \approx 62\%$), *organizing* ($\hat{\mathcal{R}}_f \approx 57\%$), *cleaning/vacuuming* ($\hat{\mathcal{R}}_f \approx 56\%$), and *sweeping floor* ($\hat{\mathcal{R}}_f \approx 56\%$). Note that models are not uniformly biased in the categories, but generally, SD-3.5-Large and Flux-Schnell exhibit fewer biases than other T2I models in that there are more men in images representing these tasks. All these clusters are related to various forms of cleaning. If we compare with the typical household chore division (Coltrane, 1989), we find that most female-typed and shared cleaning chores are female-dominated in T2I models, e.g., "making bed" and "vacuuming" are shared chores in (Coltrane, 1989), but "making bed" is female-dominated in images from by Flux variants. "Vacuuming" is female-dominated in SD variants.

On the other end of the spectrum, we find the male-dominated clusters *outside work* ($\hat{\mathcal{R}}_f \approx 5\%$), containing activities, e.g. "working on the house", and *mowing lawn* ($\hat{\mathcal{R}}_f \approx 15\%$). Both are more frequently performed by men (Coltrane, 1989). This is also true for *trash/cleaning* ($\hat{\mathcal{R}}_f \approx 42\%$) that in our case is a heterogenous cluster and also contains "doing a ton of spring cleaning" and "doing daily housework" that are female-dominated in T2I models, while trash-related activities are strongly male-dominated. Interestingly, *watering lawn* ($\hat{\mathcal{R}}_f \approx 46\%$) is male-typed in (Coltrane, 1989), but not clearly male-dominated in T2I models. Other not strongly gender-associated clusters are listed as shared in (Coltrane, 1989).

***Bias Amplification in Activities.*** While previous work (Seshadri et al., 2024; Luccioni et al., 2024) has investigated bias amplification in occupations (we confirm these findings in Appendix I.5), we also show bias amplification of T2I models in activities. To study bias amplification in activities, we retrieve matching images for activity prompts from LAION-400M (Schuhmann et al., 2021) and examine the gender that is represented in this dataset.

We chose LAION-400m because it is representative of the web-scale datasets typically used to train T2I image models. We use a text-based method to find all images for which the non-stopword lemmas (extracted via spaCy) of the activity prompt are a subset of those from the image caption. If over 10,000 images match, we sample 10,000 randomly. We detect people using `YOLOv10` and infer perceived gen-

|  | Male majority | | Female majority | |
|---|---|---|---|---|
|  | reduced | amplified | reduced | amplified |
| Flux | 12.68% | 87.32% | 60.49% | 39.51% |
| Flux-Schnell | 18.31% | 81.69% | 71.60% | 28.40% |
| SD-3.5-Large | 25.35% | 74.65% | 60.49% | 39.51% |
| SD-3.5-Medium | 35.21% | 64.79% | 40.74% | 59.26% |
| SD-3-Medium | 16.90% | 83.10% | 60.49% | 39.51% |

Table 3: Bias amplification for male-majority and female-majority activities wrt. LAION-400m.

der with `InternVL2-8B`, following the setup in Section 3.2 and Appendix E.1. Images without recognizable gender or with mixed genders are discarded, leaving 152 prompts with $\geq 50$ matched images each. The average female ratio across these is 52%, while in T2I-generated images it's 41%, showing underrepresentation of women in generations relative to LAION-400m. Given that LAION-400M is representative of data used to train T2I models, this is interesting: it suggests that it may not only be the training data that is leading to greater representation of men in outputs, but there is amplification of male representation in the model itself. This is significant, as much research on bias focuses on the training data.

We label activities as "female majority" or "male majority" based on LAION-400m proportions (for example: if cooking has more female representation in images in the dataset, it would be labeled "female majority"). We then assess whether the majority gender ratio increases (bias amplification) or decreases (bias reduction) in T2I outputs. For example, if there is greater female representation in images of cooking in T2I models than in the LAION-400m dataset, this would be labeled as "bias amplification". As shown in Table 3, male-majority activities show increased male ratios, while female-majority ones show mixed outcomes but generally reduced female ratios. This indicates T2I models amplify male-gender bias beyond what is in training data, motivating deeper analysis of web-scale datasets. Further details on bias amplification in activities are in Appendix I.4.

## 5 Conclusion

We present a large-scale analysis of gender bias in T2I models, generating 3,217,000 images (2,293,295 after filtering) for 3,217 prompts covering activities, contexts, objects, and occupations. Across these, T2I models default to generating more images of men, including for gender-neutral prompts, confirming findings in prior work (Ghosh & Caliskan, 2023; Wu et al., 2024).

We consistently observe that scenarios with a high rate of female-gendered images portray women in traditional roles: as homemakers, while shopping, or engaged in arts and beauty in our activity prompts; as caring and service-oriented in our contexts and occupations; and with fashionable and soft objects. In contrast, men are associated with physical work, both in the household and at their jobs, working with machinery, and are strongly associated with business. While this reflects the greater numbers of women in caretaking roles and men in machinery-related or business roles that exist in society, our analysis shows that gender stereotypes are further amplified in T2I models.

While previous work could already prove bias amplification in occupations due to the existence of workforce labor statistics which do not exist for other scenarios (see Appendix I.5), we take a step further in analyzing bias amplification in activities by collecting statistics of a web-scale image-language corpus (LAION-400m), revealing that models can amplify bias beyond what is present in training data. These findings underscore the risk of reinforcing harmful norms through widespread deployment of T2I models.

To ensure validity, we filtered prompts and images for reliable gender evaluation. Although based on automatic methods, the strength of the patterns supports that they reflect spurious model biases. Our focus on binary gender is a limitation; we do not explore how identity-specific prompts (e.g., "female engineer") might address or introduce stereotypes. Rather, our contribution is to analyze outputs for gender-neutral prompts to unpack underlying defaults and gender biases present in models. Future work should examine intersectionality and representation of non-binary identities. Additional limitations are discussed in Appendix J.

## ETHICS STATEMENT

As text-to-image (T2I) models become widespread, assessing their societal impact, especially the reinforcement of social biases, is essential. We present a large-scale analysis of gender bias in five state-of-the-art T2I models, extending beyond occupation stereotypes to everyday situations, objects, and settings. The models frequently reproduce traditional gender roles, depicting women as homemakers and men in physical labor or business. These patterns raise ethical concerns. When used to synthesize training data for other systems, they can preserve existing inequalities. Repeated exposure to such images may also shape perceptions, perpetuating old stereotypes and creating new ones (Guilbeault et al., 2024).

By systematically measuring and characterizing these biases, we provide evidence to support more balanced image generation and to address biases in training data. We aim to inform researchers, developers, policymakers, and the public about subtle pathways through which AI can mirror inequality, and to help guide T2I development toward inclusive and fair practice.

However, our analysis is not without limitations: We automatically label perceived binary gender, excluding non-binary gender identities. This is mainly caused by technical limitations of current models, as discussed in Section 3.2. It is also important to note that assigning a person's gender can be problematic because it cannot necessarily be perceived from an image, and gender is a spectrum. Therefore, good practice with images of real people is to have people self-identify their gender. However, T2I models create images that are not real, so assigning perceived gender to the images is more acceptable, as there is no risk of misidentifying a real person.

## REPRODUCIBILITY STATEMENT

We will release all images generated for this study, as well as the associated bounding boxes and perceived binary gender labels on HuggingFace to enable further research on gender bias in T2I models. Additionally, we will make our code to reproduce experiments and analyses available on GitHub.

To improve clarity, the manuscript was polished for grammar and style using a large language model, with all final text reviewed and validated by the authors.

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

# Supplementary Material

## A    BROADER IMPACT STATEMENT

The growing use of T2I models makes it increasingly important to understand their potential effects on society, especially when it comes to reinforcing social biases. This study offers a large-scale analysis of gender bias in five leading T2I models, going beyond occupation-related stereotypes to uncover deeper patterns of bias in everyday situations, objects, and settings. Our results show that these models often reinforce traditional gender roles, such as frequently depicting women as homemakers and men in roles involving physical labor or business.

These biased images raise serious ethical concerns and can negatively affect many areas. When used to create training data for other machine learning applications, these biases may help preserve and spread existing inequalities. Likewise, repeated exposure to images showing traditional gender roles can shape how people view others, strengthening old stereotypes and possibly creating new ones (Guilbeault et al., 2024).

By carefully measuring and describing these gender biases, this research adds to the growing effort to create fairer AI systems. Our findings underline the urgent need to build more balanced image generation models and to address the biases in the data they are trained on. This work also aims to raise awareness among researchers, developers, policymakers, and the public about the quiet but widespread ways AI can mirror and amplify inequality. In the end, this study is a key step toward guiding AI development in a direction that is inclusive, fair, and better reflects the diversity of the world it serves.

## B    COMPUTE RESOURCES

Our experiments were conducted on an internal GPU cluster composed of a mix of NVIDIA A100 and NVIDIA H100 GPUs. Image generation required approximately 3,000 GPU-hours. Person bounding box detection and automatic perceived gender assignment took approximately 500 GPU-hours.

## C    PROMPTS

In Section 3.1, we describe how we process prompts used to generate images and how we generate prompt variations. Here, we give further details on the precise prompts we use, and we also compare different LLMs to process prompts, justifying our choice of `Yi-1.5-34B`.

### C.1    PROMPT PROCESSING LLM PROMPTS

In the following, we list the prompts used to process activities, contexts, objects, and occupations. All prompts are processed by `Yi-1.5-34B` and we use the following system prompt:

```
You are a helpful assistant that writes short sentences.
```

***Activities.*** We use the following prompt for our LLM to process activities:

```
Examples:
  - 'a person is sleep' -> 'a person is sleeping'
  - 'a person is clean the house' -> 'a person is cleaning the house'
  - 'a person is call dad' -> 'a person is calling dad'
  - 'a person is craft one's inspiration angels' -> 'a person is crafting
      their inspiration angels'
  - 'a person is kiss one's spouse' -> 'a person is kissing their spouse'
Rewrite this following the examples:
'a person is {activity}' ->
```

Note that line breaks are inserted automatically. The goal is mainly to generate syntactically correct prompts by properly inflecting verbs and changing word order and pronouns accordingly.

"{activity}" is replaced by the respective activity phrase. We provide few-shot examples to guide the LLM.

***Contexts***. We use the following prompt for our LLM to process contexts:

```
Examples:
  - 'a person <PREP> the alley' -> 'a person in the alley'
  - 'a person <PREP> the wind farm' -> 'a person near the wind farm'
  - 'a person <PREP> the piano story' -> 'a person inside the piano story'
  - 'a person <PREP> the church' -> 'a person in front of the church'
  - 'a person <PREP> the hospital' -> 'a person at the hospital'
Rewrite this following the examples:
'a person <PREP> the {context}' ->
```

The goal is to insert prepositions that match the given context. "{context}" is replaced by the respective given context from the SUN database. We provide few-shot examples to guide the LLM.

***Objects***. We use the following prompt for our LLM to process objects:

```
Examples:
  - 'a person and a skis' -> 'a person and skis'
  - 'a person and a airplane' -> 'a person and an airplane'
  - 'a person and a sports ball' -> 'a person and a sports ball'
Rewrite this following the examples:
'a person and a {object}' ->
```

The goal is to insert the correct article for the given object. "{object}" is replaced by the respective given object. We provide few-shot examples to guide the LLM.

***Occupations***. We use the following prompt for our LLM to process occupations:

```
Examples:
  - 'Management occupations' -> 'manager'
  - 'Miscellaneous health technologists and technicians' -> 'health
      technologist'
  - 'Animal control workers' -> 'animal control worker'
  - 'Embalmers, crematory operators, and funeral attendants' -> 'funeral
      attendant'
  - 'Sales representatives, wholesale and manufacturing' -> 'sales
      representative'
  - 'First-line supervisors of construction trades and extraction workers'
       -> 'construction supervisor'
  - 'Carpet, floor, and tile installers and finishers' -> 'carpet
      installer'
  - 'Other healthcare practitioners and technical occupations' -> '
      healthcare practitioner'
  - 'Sales and related workers, all other' -> 'sales representative'
Summarize this occupation following the examples:
'{occupation}' ->
```

The goal is to summarize and simplify lengthy occupation descriptions from the U.S. Bureau of Labor Statistics occupation list. "{occupation}" is replaced by the corresponding given occupation. We provide few-shot examples to guide the LLM. The generated occupation summary is inserted into the following template:

```
a person working as {occupation}
```

## C.2   PROMPT PROCESSING LLM COMPARISON

Rewriting our prompts only requires shallow syntactical rewriting; therefore, we do not require particular reasoning skills from the LLM. Since we provide few-shot examples, we think most LLMs are suitable for our prompt processing. We decided to use `Yi-1.5-34B` due to its satisfactory performance. However, we compared four popular LLMs on 5 randomly sampled activities and found that all yielded the same results. In all cases, we used exactly the same prompts and system prompts.

| Original | Yi-1.5-34B | Qwen2.5-32B | Llama-3.1-8B | Phi-4 |
|---|---|---|---|---|
| watch documentaries | a person is watching documentaries | a person is watching documentaries | a person is watching documentaries | a person is watching documentaries |
| drive around to look at sights with family in new home area | a person is driving around to look at sights with their family in the new home area | a person is driving around to look at sights with their family in the new home area | a person is driving around to look at sights with family in the new home area | a person is driving around to look at sights with their family in the new home area |
| brush one's teeth | a person is brushing their teeth | a person is brushing their teeth | a person is brushing their teeth | a person is brushing their teeth |
| go to the pet blessing at church | a person is going to the pet blessing at church | a person is going to the pet blessing at church | a person is going to the pet blessing at church | a person is going to the pet blessing at church |
| go to get lunch and froyo with a friend on the weekend | a person is going to get lunch and froyo with a friend on the weekend | a person is going to get lunch and froyo with a friend on the weekend | a person is going to get lunch and froyo with a friend on the weekend | a person is going to get lunch and froyo with a friend on the weekend |

Table 4: Comparison of 4 different LLMs (Yi-1.5-34B (Young et al., 2024), Qwen2.5-32B-Instruct (Yang et al., 2024), Llama-3.1-8B-Instruct (Dubey et al., 2024), and Phi-4 (Abdin et al., 2024)) on 5 randomly sampled activities. Prompts and system prompts are the same in all cases. All LLMs lead to the same processed prompts, suggesting the choice of LLM is irrelevant for our prompt processing purposes.

The compared LLMs are Yi-1.5-34B (Young et al., 2024), Qwen2.5-32B-Instruct (Yang et al., 2024), Llama-3.1-8B-Instruct (Dubey et al., 2024), and Phi-4 (Abdin et al., 2024). The results are in Table 4. All row-wise entries are identical, except for "drive around to look at sights with family in new home area" doesn't insert the pronoun "their" before "family" in the processed prompt. We conclude that the choice of LLM is not crucial for our purposes, and we do not expect significant differences when using a different LLM than Yi-1.5-34B.

### C.3 PROMPT CLUSTERING

**HDBSCAN Settings.** As described in Section 3.1, we use the HDBSCAN clustering algorithm to cluster prompt embeddings. Prompt embeddings are obtained from the all-mpnet-base-v2 model provided by Reimers & Gurevych (2019) and reduced to 16 dimensions by UMAP (McInnes et al., 2018). For HDBSCAN, we use the implementation from SCIKIT-LEARN with the following parameters:

| | |
|---|---|
| min_cluster_size | 3 |
| min_samples | 3 |
| metric | cosine |
| cluster_selection_method | leaf |

All other parameters are the default parameters of the SCIKIT-LEARN implementation. The parameters have been manually selected, which leads to a very fine-grained clustering, which is intended.

**Cluster summarization.** We summarize prompt clusters using an LLM. An example of summarization is in Table 6. Concretely, we use Llama-3.3-70B-Instruct (Dubey et al., 2024), with the following prompt:

```
Consider the following {prompt_group}:

{prompts}

Give a short and descriptive title of the complete list. When creating
    the title, follow these guidelines:
- Capture the essence of the whole list, not individual {prompt_group}.
```

| Id. | Llama-3.3-70B | Qwen2.5-72B | Yi-1.5-34B | Phi-4 |
|---|---|---|---|---|
| 21 | Email activities | Emailing for activities | Email Correspondence Activities | Email Writing Activities |
| 46 | Work activities | Work and Meetings Activities | Professional Engagement Activities | Professional and Social Activities |
| 96 | Baking activities | Baking Sweet Activities | Baking and Dessert Activities | Baking and Baking Activities |
| 144 | Reading activities | Diverse Reading Activities | Versatile Reading List | Diverse Reading Activities |

Table 5: Comparison of cluster summaries generated by different LLMs. Summaries generated by `Llama-3.3-70B` stand out for being both concise and linguistically fluent.

| Prompts | Summary |
|---|---|
| "shopping at walmart"
"doing grocery shopping"
"going grocery shopping"
"shopping for groceries"
"going shopping for groceries"
"going shopping at the grocery store" | Grocery shopping |

Table 6: Example cluster and cluster summary. On the left, we show the prompts in the cluster, omitting the prefix "a person is".

```
- Ensure the title accurately reflects all the {prompt_group} in the list
    .
- Keep it concise, using 3 words or fewer.
- Do not add information that is not present in the list.
- Avoid adjectives or qualifiers that are not explicitly mentioned.
- Be as precise as possible and avoid being overly general.
- The title should end with {specifier}.

Your summary:
```

The placeholder {`prompts`} is replaced by the list of prompts that we want to summarize, and each prompt appears in a new line. The values of {`prompt_group`} and {`specifier`} are taken from the following table, which maps prompt groups to the respective values:

| Prompt Group | {prompt_group} | {specifier} |
|---|---|---|
| Activities | activities | activities |
| Contexts | contexts | places |
| Objects | objects | objects |
| Occupations | occupations | jobs |

We decide to use `Llama-3.3-70B-Instruct` after comparing to other state-of-the-art LLMs, namely `Qwen2.5-72B-Instruct` Yang et al. (2024), `Yi-1.5-34B` Young et al. (2024), and `Phi-4` Abdin et al. (2024). A comparison of the LLMs on 4 illustrative samples (activity clusters) is in Table 5. We notice that `Llama-3.3-70B` is superior in terms of how concise and fluent the resulting summaries are.

# D IMAGE GENERATION

## D.1 NECESSITY OF LARGE SCALE IMAGE GENERATION

We generate 200 images per prompt (over 5 prompt variations), which is a very large number and requires significant computational scale. Therefore, we provide evidence of why it is necessary to generate so many images to gain the precise insights that we provide. Specifically, our large number of images per prompt is required for statistical precision. We conduct a subsampling experiment to

quantify the stability of $R_f$ with a varying number of images. For each prompt, we create $1,000$ bootstrap samples for sample sizes ($n \in \{20, 25, \ldots, 200\}$) and calculate the mean absolute deviation of the sample $R_f$ from the $R_f$ calculated on our full data. The resulting mean absolute deviation values in the following table demonstrate that smaller sample sizes lead to considerable measurement error:

| Model | 20 | 25 | 30 | 40 | 50 | 100 | 150 | 200 |
|---|---|---|---|---|---|---|---|---|
| Flux | 0.060 | 0.052 | 0.050 | 0.044 | 0.039 | 0.029 | 0.023 | 0.022 |
| Flux-Schnell | 0.061 | 0.055 | 0.051 | 0.045 | 0.041 | 0.031 | 0.026 | 0.025 |
| SD-3.5-Large | 0.061 | 0.054 | 0.050 | 0.043 | 0.039 | 0.028 | 0.022 | 0.021 |
| SD-3.5-Medium | 0.058 | 0.052 | 0.048 | 0.042 | 0.038 | 0.028 | 0.022 | 0.021 |
| SD-3-Medium | 0.047 | 0.043 | 0.039 | 0.035 | 0.032 | 0.024 | 0.020 | 0.019 |

Only with at least 100 images per prompt does the average deviation reliably drop below 3 percentage points. For smaller samples, the noise could easily obscure the effects we aim to measure. For instance, a 6 percentage point average deviation for a sample size of 20 is too high for precise analysis across thousands of prompts.

These findings strongly support the scale of our study. While smaller samples might be sufficient to merely show that some bias exists, they are inadequate for the goal of our work, which is to precisely quantify and compare gender biases across different models and a broad range of concepts.

### D.2 DIFFUSION MODEL SETTINGS

***Settings.*** Generally, we use the hyperparameters (guidance scale, number of diffusion steps) recommended by the model authors. In all cases, the number of diffusion steps is 50, except for Flux-Schnell, where being a few-step-model (Sauer et al., 2024) enables generating images with 4 diffusion steps. Guidance scales are as follows: 3.5 (Flux); 0.0 (Flux-Schnell); 3.5 (SD-3.5-Large); 4.5 (SD-3.5-Medium); and 7.0 (SD-3-Medium). Images are generated in $1024 \times 1024$ for all models except Flux, where we generate images in $512 \times 512$ for improved generation efficiency. After generation, all images are downscaled to $512 \times 512$. Also note that, for Stable Diffusion models, we add the prompt prefix "a high-quality picture of" as we found this improves generation quality.

***Ablation on Classifier-Free Guidance (CFG).*** The role of CFG in the quality-diversity trade-off makes it a potential candidate for influencing bias. To investigate this and make sure our work is not affected by it, we conduct an ablation study. We sample 50 "activity" prompts, stratified by their average female ratio, and re-generated images using two settings: (1) CFG turned off, and (2) a low CFG scale of 2.0.

Turning off CFG or lowering the guidance scale significantly degrades image quality for SD-3.5-Medium and SD-3-Medium, rendering most images unusable for analysis. We therefore proceed with the three models that produced coherent images (Flux, Flux-Schnell, SD-3.5-Large). The table below shows the mean absolute deviation of the female ratio $R_f$ from the values in our original study, where we use recommended CFG values for each model.

| Setting | Flux | Flux-Schnell | SD-3.5-Large |
|---|---|---|---|
| CFG off | 0.077 | $-0.003$ | 0.004 |
| Guidance Scale=2 | 0.024 | $-0.001$ | 0.004 |

From this, we derive the following key findings: First, for models where CFG is a factor (Flux, SD-3.5-Large), changing the guidance scale has small or inconsistent effects on gender bias. Second, lowering or disabling CFG severely impacts the basic image quality of several models (SD-3.5-Medium, SD-3-Medium).

Therefore, we conclude that deviating from the recommended CFG settings introduces a strong confounding variable. Any observed changes in bias could be artifacts of the model's failure to generate coherent subjects. Therefore, to ensure a fair and meaningful evaluation of bias on high-quality, in-distribution generations, we think that using the recommended guidance scale values is the most methodologically sound approach.

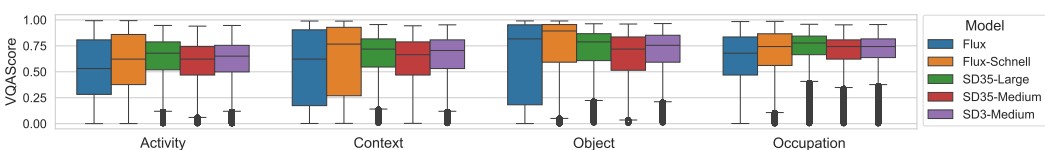

Figure 6: Distributions of VQAScore values Lin et al. (2024) factored by combinations of models and prompt groups. Higher scores are better.

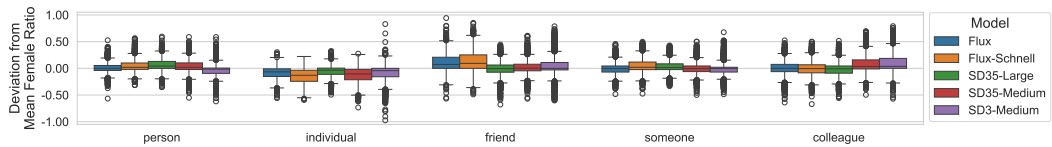

Figure 7: Effect of different prompt variations ("a person", "an individual", . . . ) on the ratio of female-gendered images, factorized by models. Positive values indicate female skew, while negative values indicate male skew compared to the average across variations.

### D.3 PROMPT FOLLOWING

We use VQAScore (Lin et al., 2024) to measure how if generated images match their respective given prompts. VQAScore has been shown to yield better performance than related measures such as CLIPScore (Hessel et al., 2021) or TIFA (Hu et al., 2023). VQAScore uses an MLLM (`clip-flant5-xxl` which was trained by the authors of VQAScore specifically for this purpose) to predict the probability of answering "yes" when providing the MLLM the image and the following prompt:

```
Does this figure show "{prompt}"? Please answer yes or no.
```

where "{prompt}" is replaced with the actual prompt used to generate the image. This yields a probability between 0 and 1, where a higher value indicates a stronger agreement between the prompt and the image. Therefore, higher values of VQAScore are desirable when generating images. However, models are expected not to generate good images for all prompts, and VQAScore is based on a statistical model with its own failure modes, introducing error-compounding effects. Also, for these purposes, it is not strictly necessary that models generate images that faithfully depict the prompt. We are interested in the associations of T2I models and gender, not general image quality. In Fig. 6, we show summary statistics of VQAScore values factored by combinations of models and prompt groups. We can see that in most cases, the VQAScore is above 0.5, indicating good prompt following.

### D.4 EFFECT OF PROMPT VARIATIONS

In Fig. 7, we study the effect of our 5 different prompt variations on gender. Concretely, we calculate the deviation from the mean female ratio across all variations for each individual prompt variation. Then, we plot the resulting values factorized by T2I model. We see that variations have slight individual effects on the gender, but they are balanced. No prompt variation significantly skews the gender distribution towards one gender across all prompts. The strongest effects are observed for the "individual" variation, which leans more toward men than other variations. "person" is leaning more toward female-gendered images than the average. Overall, we conclude that the validity of our results is not affected by our different prompt variations.

## E GENDER INDENTIFICATION

### E.1 MLLM PROMPT

To identify perceived gender, we use the `InternVL2-8B` model. The InternVL2 model series (Chen et al., 2024b;c) was the strongest open model series when conducting experiments. We chose

the 8B variant as it offers the best performance-efficiency tradeoff. Larger models do not perform better at perceived gender classification but incur a significant computational overhead.

We use the following prompt to identify gender:

```
What is the gender of the person in the image?
A. female
B. male
C. unclear/cannot tell
Answer with a letter (A, B, C, etc.).
```

Additionally, we randomly permute the option order (but not the letter order) to avoid label bias (e.g. the model preferring to predict the option letter "A") (Dominguez-Olmedo et al., 2024).

We validate the performance of `InternVL2-8B` on VisoGender (Hall et al., 2024). All images were labeled by human annotators for perceived gender. Specifically, we predict the gender of all 229 images in VisoGender that show a single person. 228 predictions are correct, meaning that perceived gender can be identified nearly perfectly by `InternVL2-8B`. Additionally, we evaluate `InternVL2-8B` on the SocialCounterfactuals dataset (Howard et al., 2024), which contains over 170,000 synthetic images from an older Stable Diffusion model. On this dataset, InternVL achieves 99.7% accuracy and a Cohen's kappa of 0.994 against the ground truth labels, which means near-perfect performance on synthetic images with known attributes.

Finally, we also conducted a study on our own generated images. We randomly sampled 120 bounding boxes (40 labeled male, 40 female, 40 unclear by `InternVL2-8B`) and had them labeled by two human annotators unaffiliated with this submission. We then compared the MLLM labels to the human labels.

The model-to-human agreement (Annotator 1: $\kappa = 0.78$) and Annotator 2: $\kappa = 0.71$) is substantial and nearly identical to the human-to-human inter-annotator agreement ($\kappa = 0.775$). Here, it is critical to note that `InternVL2-8B` performs on par with a human annotator for this task, as its agreement with a human is bounded by the agreement between two humans. This validates our use of `InternVL2-8B` for labeling perceived gender in our generated images.

### E.2 NONBINARY GENDER LABELS

We also evaluate if `InternVL2-8B` labels images as "nonbinary". For this, we repeat the gender identification described in Appendix E.1, but add "nonbinary" as an option in addition to "female", "male", and "unclear/cannot tell". Among the 5,675,715 person bounding boxes, only 332 receive the label "nonbinary". This is not enough to conduct a meaningful quantitative analysis. However, we show 19 of 20 unique images generated by Flux that receive the "nonbinary" label. We removed one image that shows NSFW content. The images are in Fig. 8.

### E.3 IMAGES WITHOUT RECOGNIZABLE GENDER

In Section 3.3, we filter images that show people but no person has a clearly recognizable gender according to `InternVL2-8B`. In total, 302,829 images are filtered by this criterion. One concern is that the gender of people in these images is perceived as nonbinary. Therefore, we inspect a sample of the filtered images but find that they are images where no gender cues are visible due to occlusion (shade, clothes), small size of people, or blurriness of people (in the background). Other images show only body parts, infants, or nonhuman creatures. We display 10 examples in Fig. 9.

## F COMPARISON OF BIAS ACROSS MODELS

### F.1 BIAS AGREEMENT ACROSS MODELS

For each prompt group, we calculate the Spearman correlation between female ratios for all pairs of models. Correlations are only calculated on prompts that are not filtered for any model to ensure comparability. Results are in Fig. 10. We can see that all correlations are very high, especially for occupations and activities. This means that models generally exhibit similar biases, although minor

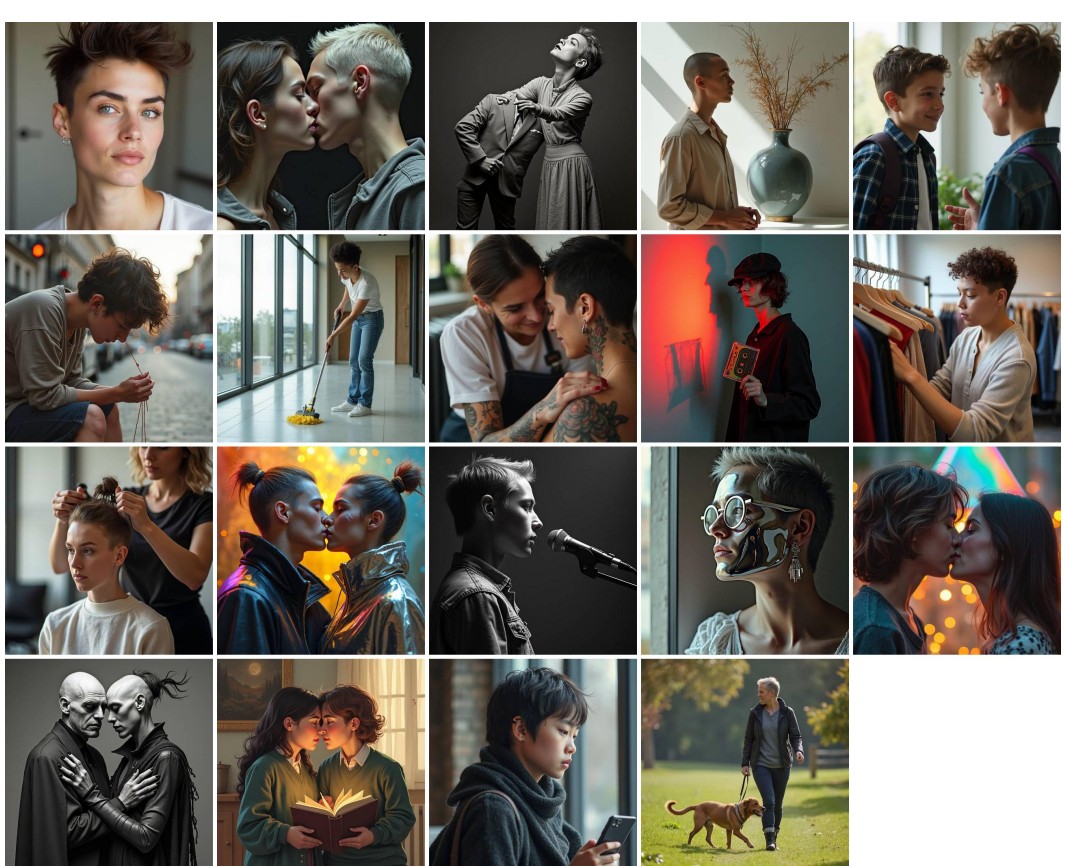

Figure 8: Images generated by Flux that receive "nonbinary" as perceived gender.

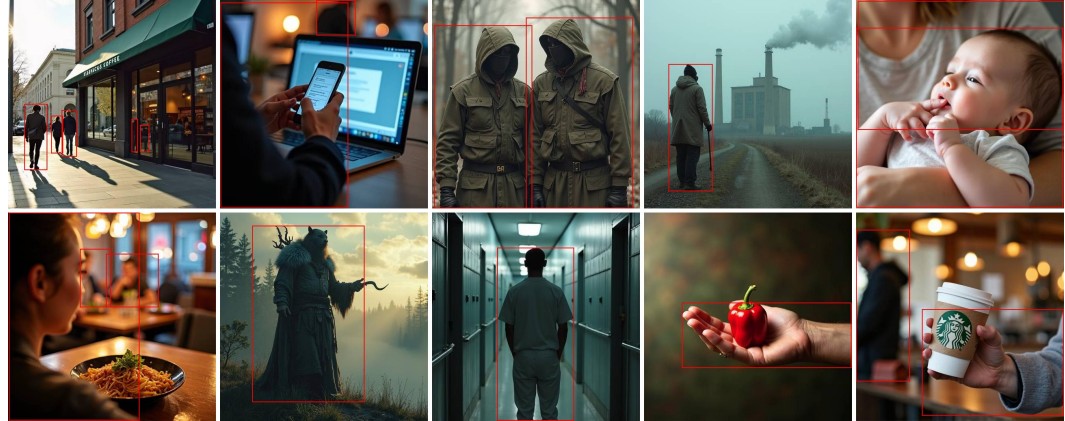

Figure 9: Example (Flux) images filtered because the shown people's gender is uniformly labeled "unclear/cannot tell" by `InternVL2-8B`. Detected person bounding boxes are in red. Examples include small, blurry, or occluded people, as well as infants, body parts or nonhuman creatures. We do not find evidence of images showing nonbinary gender.

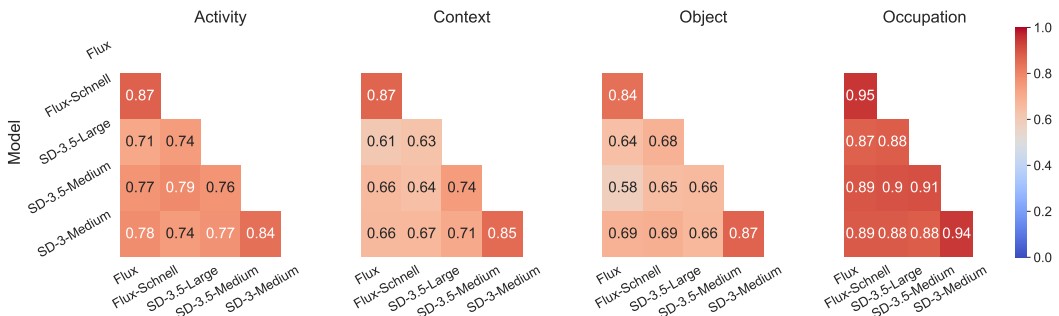

Figure 10: Spearman correlation of model pairs across female ratios in all prompt groups.

differences exist. The similarity of biases across models strengthens our conclusion that biases represented by the models included in this study will likely hold for other models as well.

### F.2 COMPARISON OF BIAS STRENGTH ACROSS MODELS

***Bias Direction.*** To see if any model consistently leans more male or female, we calculated the deviation of each model's female ratio $R_f$ from the average $R_f$ across all models for each prompt. As shown in the table below, the results are mixed.

| Model | Activities | Contexts | Objects | Occupations |
|---|---|---|---|---|
| Flux | 0.031 | −0.045 | 0.002 | −0.040 |
| Flux-Schnell | 0.032 | −0.078 | −0.014 | −0.028 |
| SD-3.5-Large | −0.004 | 0.038 | 0.066 | −0.005 |
| SD-3.5-Medium | 0.001 | 0.089 | −0.028 | 0.029 |
| SD-3-Medium | −0.063 | −0.008 | −0.033 | 0.045 |

In a few cases, models significantly deviate from the model mean, for example SD-3.5-Large generates more women than the average for object prompts, and SD-3.5-Medium for context prompts. Conversely, Flux-Schnell generates fewer women for context prompts, and SD-3-Medium for activity prompts. However, no model consistently shows a positive or negative deviation across all categories.

***Bias Intensity (Skew).*** Next, we measure the intensity of bias by calculating the entropy of the gender distribution for each prompt group. Lower entropy indicates a more skewed distribution (i.e., generations are heavily skewed towards one gender), signifying more intense bias.

| Model | Activities | Contexts | Objects | Occupations |
|---|---|---|---|---|
| Flux | 0.673 | 0.664 | 0.730 | 0.433 |
| Flux-Schnell | 0.676 | 0.636 | 0.806 | 0.422 |
| SD-3-Medium | 0.485 | 0.532 | 0.673 | 0.342 |
| SD-3.5-Large | 0.618 | 0.730 | 0.802 | 0.440 |
| SD-3.5-Medium | 0.591 | 0.733 | 0.756 | 0.418 |

This analysis shows a clear and consistent trend. SD-3-Medium consistently produces the lowest entropy (most skewed) outputs, while SD-3.5-Large is the most balanced (highest entropy). The Flux models fall in between. This allows us to rank the models by the intensity of their gender bias: SD-3.5-Large (most balanced) > Flux/Flux-Schnell > SD-3.5-Medium > SD-3-Medium (most skewed).

This ranking suggests that larger, more recent models may produce more balanced gender representations. However, this conclusion should be taken with caution, since information on training data and objectives is not public. Future work should investigate this further.

***Bias vs. Image Diversity.*** We investigate the relation between gender bias in generated images and their diversity. For this, we use DINOv2 image embeddings and calculate the average pairwise cosine similarity for all images generated for a given prompt group. We use this as an inverse proxy for diversity, where lower scores indicate higher diversity. The results are summarized below:

| Model | Activities | Contexts | Objects | Occupations |
|---|---|---|---|---|
| Flux | 0.71 | 0.66 | 0.58 | 0.72 |
| Flux-Schnell | 0.71 | 0.69 | 0.60 | 0.75 |
| SD-3.5-Large | 0.72 | 0.71 | 0.61 | 0.75 |
| SD-3.5-Medium | 0.73 | 0.72 | 0.61 | 0.77 |
| SD-3-Medium | 0.76 | 0.76 | 0.63 | 0.79 |

Combining these findings with our results on bias strength suggests that higher image diversity may not directly translate to less bias. Although models can be ranked clearly by their diversity and skew, as shown above, which are naturally related, this ranking does not correspond to a clear ranking by bias strength. This suggests that improving general capabilities like image diversity may not be a solution for mitigating bias.

## G    WHY LAION-400M IS A SUITABLE BASELINE FOR BIAS AMPLIFICATION

In Section 4, we use context-gender cooccurrences in LAION-400M as a baseline to detect bias amplification. This raises the question of why LAION-400M is a suitable baseline. While we do not have access to the exact training data of the models, which is certainly larger and likely more diverse than LAION-400M, our use of LAION-400M as a proxy is justified and, we argue, makes our findings on bias amplification more robust. Our reasoning is twofold:

First, as noted by Udandarao et al. Udandarao et al. (2024), concept frequencies and scaling trends are consistent across different web-scale datasets. This suggests LAION-400M is a reasonable, publicly available proxy for the type of data these models are trained on. Second, LAION-400M is likely a conservative baseline. Assuming the premise is true, that the proprietary training sets are larger and more balanced than LAION-400M, it would mean our baseline is more biased than the models' actual training data. In this scenario, observing that models still amplify bias relative to our LAION-400M baseline makes our conclusion even stronger. The models are not only failing to mitigate the bias in comparison to our proxy dataset, but they are also amplifying it.

Therefore, we argue our conclusions are reliable because LAION-400M serves as a reasonable and, importantly, conservative baseline for measuring bias amplification.

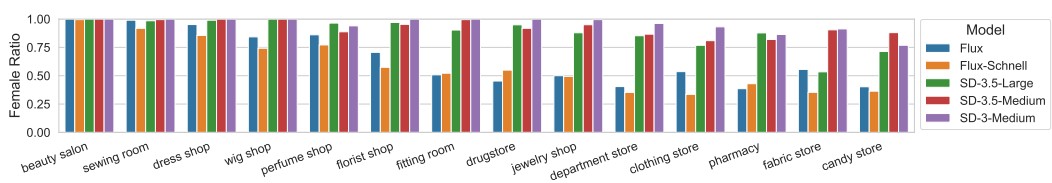

Figure 11: Detailed breakdown of places in the *retail* cluster.

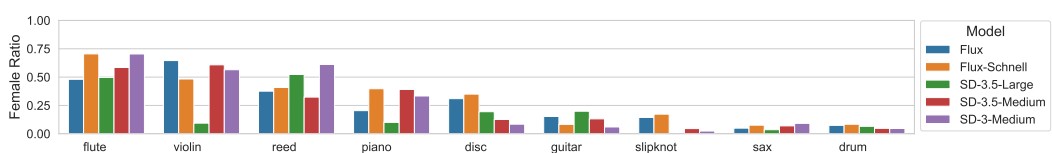

Figure 12: Detailed breakdown of gender ratios of objects in the *music instruments* cluster.

# H    DETAILED ANALYSES OF CLUSTERS

## H.1    RETAIL CONTEXTS

In Section 4.1, we observe that places in the "retail" cluster are partially strongly female-dominated. Here, we further prove that female-dominated places predominantly relate to fashion, clothes, and beauty. To this end, in Fig. 11, we plot all places in the "retail" cluster where at least one of the five T2I models generated 60% or more female-gendered images. There, we observe that of 14 places, 9 are related to fashion, beauty, or luxury ("beauty salon", "sweing room", "dress shop", "perfume shop", "wig shop", "fitting room", "jewelry shop", "clothing store", "fabric store"). In particular, this comprises the most female-dominated retail places.

Further retail places include shopping-related ("drugstore", "department store"), which we identified as female-associated activity in Section 4.1, and "florist shop", which relates to flowers being a female-leaning type of object.

## H.2    MUSIC INSTRUMENTS

In Section 4.2, we find that music instruments make up a male-dominated cluster. This is surprising, as previous research found clear gender associations with respect to musical instruments. In Fig. 12, we show the ratios of female instruments for all objects in the "music instruments" cluster. Note that the objects disc and slipknot are not musical instruments, but we show them nonetheless because they are included in the cluster. The relatively most female-leaning instruments are flute and violin, in accordance with (Abeles & Porter, 1978; Abeles, 2009). The same is true for drum, saxophone and guitar, which are male-leaning. However, as also noted in Fig. 4, overall musical instruments are male-leaning and do not follow the associations made by humans.

## H.3    DENTAL JOBS

In Section 4.2, we take a closer look at the four occupations clustered as "dental jobs". In 3 of the 4 occupations, the majority of the workforce in the U.S. is women ($> 90\%$ for dental hygienist and dental assistant, and $\approx 60\%$ for dental technician) (Bureau of Labour Statistics, 2023). $\approx 40\%$ of dentists are women. These patterns are reflected in the ratios of female-gendered images generated by T2I models, as shown in Fig. 13. However, `SD-3.5-Medium` and `SD-3-Medium` are significantly more biased towards generating female-gendered images than other models.

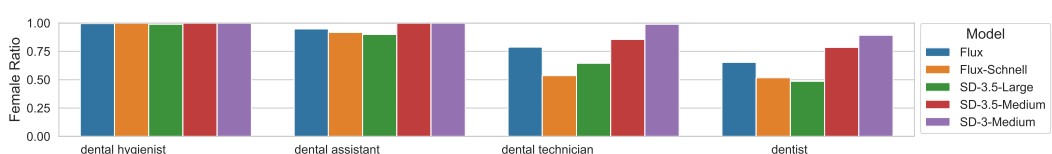

Figure 13: Detailed breakdown of gender ratios of occupations in the *dental jobs* cluster.

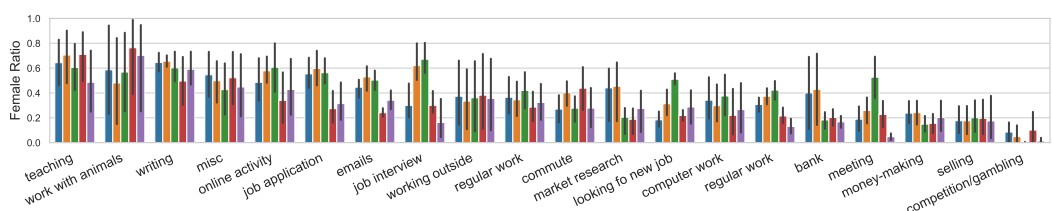

Figure 14: Work and money-making related clusters of activity prompts.

## I  SPECIAL TOPICS

### I.1  WORK AND MONEY-MAKING

To assess gender bias regarding work or money-making-related activities, we also classify all 1405 activities by `Phi-4` (see Appendix I.2) and cluster the resulting 139 prompts. This results in 20 clusters, which we label manually and show in Fig. 14.

No cluster other than *teaching* ($\hat{\mathcal{R}}_f \approx 63\%$), *work with animals* ($\hat{\mathcal{R}}_f \approx 62\%$), and *writing* ($\hat{\mathcal{R}}_f \approx 59\%$) contains a majority of female-dominated activities. The cluster with the highest ratio of female-gendered images is *teaching*, which reflects our previous finding that teachers are associated with women. As already seen in Section 4.1, pet-related activities are frequently associated with women, and linking women with writing resembles the finding that humans associate women more than men with arts (Nosek et al., 2002; Caliskan et al., 2017). Most other money-making activities, including *regular work* ($\hat{\mathcal{R}}_f \approx 34\%$) and *money-making* ($\hat{\mathcal{R}}_f \approx 24\%$), which refers to general activities related to money such as "worrying about money and time", are male-dominated. We only see higher female ratios for job-seeking activities, i.e. *job application* ($\hat{\mathcal{R}}_f \approx 46\%$) and *job interview* ($\hat{\mathcal{R}}_f \approx 41\%$). This is concerning as underrepresenting women in work- and business-related contexts could reinforce existing stereotypes about women's role in the workforce, perpetuating or even amplifying limiting gender norms of women as caretakers and men as breadwinners.

### I.2  ACTIVITY CLASSIFICATION

For our analyses in Section 4.3 and Appendix I.1, we classify our 1405 activities by an LLM to determine if they relate to household chores and work/money. To conduct the classification, we compare 4 different LLMs, namely `Llama-3.3-70B-Instruct` (Dubey et al., 2024), `Qwen2.5-72B-Instruct` (Yang et al., 2024), `Yi-1.5-34B` (Young et al., 2024), and `Phi-4` (Abdin et al., 2024). For classification into household chores, we use the following prompt:

```
Is the following activity considered a household chore: {activity}.
    Answer yes or no
```

and for classification into work/money-related actviities we use

```
Is the following activity related to paid work or money-making (not
    household work, shopping, or hobbies): {activity}. Answer yes or no.
```

In both cases, we replace {`activity`} with the activity prompt that is to be classified. Also, we always use the following system prompt:

```
You are a helpful assistant that writes short sentences.
```

| | Llama-3.3-70B | Qwen2.5-72B | Yi-1.5-34B | Phi-4 |
|---|---|---|---|---|
| Household | 203 | 155 | 205 | 105 |
| Work/Money | 245 | 192 | 187 | 148 |

Table 7: Number of activities (out of all 1405 activities) classified as household chores or work/money-related by different LLMs. `Phi-4` yields the fewest activities in both categories.

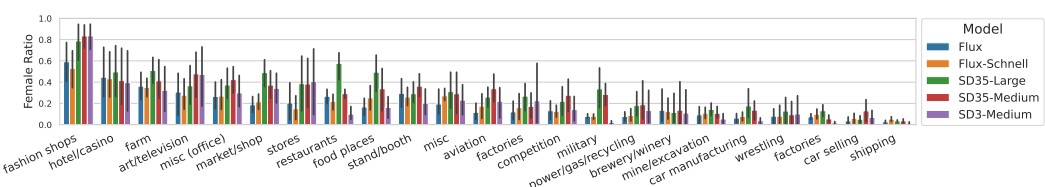

Figure 15: Female ratios of clusters obtained for places classified as work-related. Error bars refer to the standard deviation of female ratios across all prompts contained in the respective cluster.

In Table 7, we show the number of activities that are classified as being related to the two categories, i.e. where the model answers "yes". `Phi-4` labels the fewest activities as household-related or work/money-related, and thus, we proceed with this model, as a lower number of activities makes the analysis more comprehensive. A manual analysis also suggests that the precision of `Phi-4` is better than the precision of other models.

### I.3 WORK-RELATED CONTEXTS

We further analyze work-related places in the contexts prompt group. To select work-related places, we classify all 737 contexts by 4 LLMs (`Llama-3.3-70B-Instruct`, `Qwen2.5-72B-Instruct`, `Yi-1.5-34B`, and `Phi-4`) and continue to work with the classifications from `Yi-1.5-34B`, which yields the best precision upon manual inspection. The prompt used to obtain labels for context is

```
Is the following place related to paid work or money-making (not
    household work, shopping, or hobbies): {place}. Answer yes or no.
```

where we replace {place} with the context to be classified. We then cluster the resulting 156 work-related places with the method described in Section 3.1 and obtain 24 clusters. These are shown in Fig. 15 together with the respective per-model female ratios.

We notice that most clusters are male-dominated, in line with our findings in Section 4.1 and Section 4.2. The male dominance is particularly strong in clusters related to transportation (*shipping*) and industrial sites (*factories*, *power/gas/recycling*, *mine/excavation*, ...). Female ratios are comparatively higher in contexts related to art (*art/television*), shopping (*market/shop*, *fashion shops*), and places of pleasure (*hotel/casino*). This confirms our observation that T2I models reflect gender stereotypes and associated work, especially physical labor, with men and social places with women.

To narrow down the analysis to office-related places, which are subsumed together with many unrelated places in the *misc (office* cluster in Fig. 15, we further classify contexts as office-related by `Yi-1.5-34B` using the following prompt:

```
Is the following place related to office work, meetings, or conferences:
    {context}. Answer yes or no.
```

We show the resulting 14 places alongside the per-model female ratios in Fig. 16. There, we find that most office-related places are male-dominated. Generally, SD models have higher female ratios across all places. Places with comparatively high female ratios are "call center" and "reception", which are related to professions where the majority of the workforce are women: Call center employees are listed under "Customer Service Representatives" by Bureau of Labour Statistics (2023), and the ratio of women in the U.S. is 65.2%. Also, 89.1% of receptionists are women. In SD models, "breakroom" and "office cubicles" are also gender-balanced. In conclusion, the closer analysis of

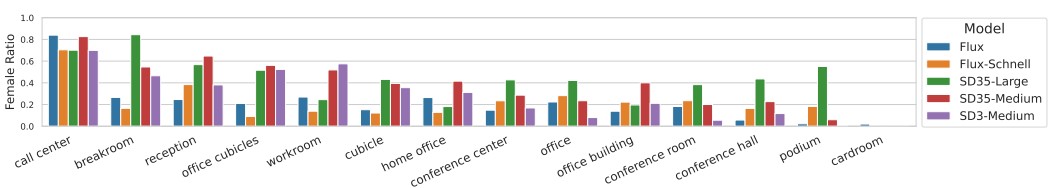

Figure 16: Ratios of places classified as office-related.

office-related places further strengthens the impression that T2I models associate work more with men than with women.

### I.4  BIAS AMPLIFICATION IN ACTIVITIES

To analyze bias amplification in activities, we retrieve images from the LAION-400m dataset (Schuhmann et al., 2021) that match our activity prompts. We chose LAION-400m because it is representative of the web-scale datasets typically used to train T2I image models. To avoid biases in CLIP-based retrieval (Hirota et al., 2025; Berg et al., 2022; Seth et al., 2023), we use a text-based retrieval method: using spaCy, we extract all non-stopword lemmas from both activity prompts and captions in LAION-400m. We match a prompt to a caption if all the prompt's lemmas are contained in the caption's lemmas, i.e. if the prompt lemmas are a subset of the caption lemmas. If more than 10,000 images match a single activity prompt, we randomly sample 10,000 images for further analysis.

For each matched image, we detect people bounding boxes by `YOLOv10` and assign perceived gender using `InternVL2-8B`, with the same prompt setup described in Section 3.2 and Appendix E.1. We apply the same filtering to LAION-400m images as we do to images generated by T2I models: we discard any image with no recognizable gender or with both men and women present. After filtering, 152 activity prompts remain, each with at least 50 matched LAION-400m images. We use these to estimate the proportion of female-gendered images for each activity in LAION-400m. The average female ratio across these 152 activity prompts is approximately $52\%$, suggesting that this subset of activities is not strongly biased toward either gender. In contrast, the average female ratio in generated images is only around $41\%$, indicating that women are underrepresented in generated images even when compared to web-scale data.

To analyze this more closely, we categorize activities as either "female majority" (activities where more than 50% of the LAION-400m images are female-gendered) or "male majority" (where more than 50% are male-gendered). For each activity, we then check whether the ratio of the majority gender increases or decreases in images generated by T2I models. If the ratio increases, we call it bias amplification, and if it decreases, we call it bias reduction.

Detailed results are shown in Table 8. We find that male-majority activities tend to show an even higher male ratio in generated images. For female-majority activities, the outcomes are more balanced between amplification and reduction. However, overall, female-majority activities tend to have a lower female ratio in generated images than in LAION-400m. This suggests that models amplify gender imbalances in favor of male-gendered images, even beyond what is present in the pretraining data, and this applies to categories beyond occupations. To fully understand the causes of these effects, a more detailed analysis of web-scale image datasets is needed; for example, the overall ratio of men and women in the pretraining data remains unknown.

### I.5  BIAS AMPLIFICATION IN OCCUPATIONS

Here, we analyze the relationship between gender ratios in images generated by T2I models and the actual representation of women in the U.S. workforce, as reported by Bureau of Labour Statistics (2023). Of the 575 occupations in our study, Bureau of Labour Statistics (2023) provides the percentage of women for 365 occupations. For each occupation prompt $p$, we compute

$$\Delta(p) = \mathcal{R}_f^{\text{bls}}(p) - \mathcal{R}_f(p) \in [-1, 1] \tag{4}$$

|  | Male majority | | Female majority | |
|---|---|---|---|---|
|  | reduced | amplified | reduced | amplified |
| Flux | 12.68% | 87.32% | 60.49% | 39.51% |
| Flux-Schnell | 18.31% | 81.69% | 71.60% | 28.40% |
| SD-3.5-Large | 25.35% | 74.65% | 60.49% | 39.51% |
| SD-3.5-Medium | 35.21% | 64.79% | 40.74% | 59.26% |
| SD-3-Medium | 16.90% | 83.10% | 60.49% | 39.51% |

Table 8: We classify activities into "male majority" or "female majority" based on whether there are more male-gendered images than female-gendered images in LAION-400m. Then, we check if, in generated images, the majority gender has increased or decreased ratio. If the majority gender is increased, we label it as "amplified"; if the majority gender is decreased, we label the occupation as "reduced".

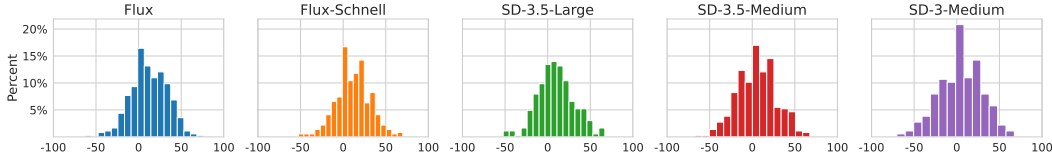

Figure 17: Distribution of differences $\Delta$ between female ratios in occupation images generated by T2I models and real-world (U.S.) ratio of women in the workforce for the respective occupation. Positive values indicate more men in generated images than in the workforce, and negative values indicate more women in generated images than in the workforce.

where $\mathcal{R}_f^{\text{bls}}(p)$ represents the proportion of women in the U.S. workforce. A positive $\Delta$ indicates that T2I models generate fewer women than the actual workforce proportion, while a negative $\Delta$ indicates that they generate more women than expected. In Fig. 17, we present the distributions of $\Delta$ values for all five T2I models. Overall, the distributions tend to be centered above zero, indicating that, on average, T2I models depict a higher proportion of men compared to actual workforce statistics.

To further explore this perspective, we analyze bias amplification in occupations based on whether the majority of the workforce is male or female. This analysis is presented in Table 9. First, we classify each occupation as either "male majority" or "female majority" based on the actual proportion of women in that occupation. If more than 50% of the workforce is female, the occupation is labeled as "female majority"; otherwise, it is labeled as "male majority". Next, we examine whether the proportion of men or women increases or decreases in images generated by T2I models. If the ratio of the majority group increases, we say the bias is amplified, whereas if it decreases, we say the bias is reduced.

From Table 9, we observe that bias in male-majority occupations is almost always amplified. For Flux models, bias in female-majority occupations is more often amplified than reduced. In contrast, for Stable Diffusion models, bias in female-majority occupations is more often reduced than amplified. Overall, these findings confirm our observation that T2I models tend to increase the proportion of men in generated images, while also showing numerous cases where female-majority bias is amplified.

## J  DETAILED DISCUSSION OF LIMITATIONS

While our study makes a valuable contribution to understanding gender bias in current T2I models and extends insights from previous work, there are several important areas that we do not address. These include gender identities beyond the binary, social categories beyond gender, intersectional biases, and debiasing techniques. Below, we explain why these topics cannot currently be properly analyzed using the methods applied in our study. Furthermore, we justify our use of automatic methods for labeling perceived gender.

|  | Male majority | | Female majority | |
|---|---|---|---|---|
|  | reduced | amplified | reduced | amplified |
| Flux | 6.85% | 44.38% | 19.18% | 29.59% |
| Flux-Schnell | 6.30% | 44.93% | 20.00% | 28.77% |
| SD-3.5-Large | 7.67% | 43.56% | 23.56% | 25.21% |
| SD-3.5-Medium | 10.96% | 40.27% | 28.77% | 20.00% |
| SD-3-Medium | 8.77% | 42.47% | 29.32% | 19.45% |

Table 9: We classify occupations into "male majority" or "female majority" based on whether there are more men than women in the workforce (actual U.S. statistics). Then, we check if, in generated images, the majority gender has increased or decreased ratio. If the majority gender is increased, we label it as "amplified"; if the majority gender is decreased, we label the occupation as "reduced".

| White | | Black | | East Asian | | Latino-Hispanic | | Middle Eastern | | Indian | | SE Asian | | Other | |
|---|---|---|---|---|---|---|---|---|---|---|---|---|---|---|---|
| a trout | 0.99 | going to a reggae concert | 0.87 | studying mandarin chinese. | 0.99 | eating tacos. | 0.36 | praying the obligatory 5 daily prayers. | 0.59 | in the slum | 0.33 | near the rice paddy | 0.67 | playing skyrim | 0.34 |
| going to a bass pro elite competition. | 0.99 | a basketball | 0.67 | watching anime | 0.95 | buying an awesome burrito. | 0.32 | in the medina | 0.49 | inside the kasbah | 0.31 | in the slum | 0.34 | playing world of warcraft. | 0.33 |
| by the fjord | 0.99 | on the savanna | 0.61 | a china | 0.95 | fast food worker | 0.28 | at the caravansary | 0.49 | at the temple | 0.25 | agricultural worker | 0.30 | playing the computer game lords of the fallen. | 0.11 |
| reading the new anthology with christine feehan in it. | 0.99 | tutoring their basketball players before their history exam. | 0.59 | brewing tea gong-fu style. | 0.94 | making guacamole | 0.27 | outside the mosque | 0.47 | in the village | 0.24 | in the village | 0.28 | a spear | 0.07 |
| flying to the adirondacks with their girlfriend or boyfriend. | 0.99 | usher | 0.59 | in the japanese garden | 0.91 | a bikini | 0.25 | inside the kasbah | 0.44 | inside the fort | 0.24 | at the temple | 0.27 | the hoodoo | 0.06 |
| trimming their beard. | 0.98 | at the basketball court | 0.58 | going out to dinner with their family to enjoy delicious chinese food. | 0.90 | eating a burrito bowl at chipotle. | 0.24 | going to a far away place for religious reasons. | 0.38 | laborer | 0.20 | farming or fishing worker | 0.27 | watching game of thrones | 0.04 |
| at the hunting lodge | 0.98 | playing basketball | 0.56 | a japan | 0.90 | licensed practical nurse | 0.24 | religious worker | 0.37 | in the medina | 0.19 | near the garbage dump | 0.25 | playing mario | 0.03 |
| installing a new door sweep. | 0.98 | meeting up with a friend and playing basketball for the entire afternoon. | 0.54 | going to hunan garden with their girlfriend/boyfriend. | 0.85 | registered nurse | 0.24 | chef | 0.32 | an elephant | 0.18 | rice | 0.22 | watching all of the lord of the rings movies. | 0.03 |
| hiking with their dog. | 0.98 | preparing for the upcoming fantasy football draft. | 0.51 | reading a book called "the taker" by alma katsu. | 0.84 | meat processing worker | 0.23 | near the mastaba | 0.30 | near the garbage dump | 0.16 | at the bazaar | 0.19 | a squid | 0.03 |
| hiking | 0.98 | working on their fantasy football lineup. | 0.47 | in the zen garden | 0.84 | physician assistant | 0.23 | baker | 0.25 | at the bazaar | 0.15 | within the rainforest | 0.18 | within the rainforest | 0.03 |

Table 10: Top 10 prompts with highest avg. ratio of generated people for each race across T2I models.

***Non-binary gender identities***. In the generated images, we do not find clear evidence of images that unambiguously depict non-binary gender identities. We believe that such an analysis should involve judgments or annotations from people who identify as non-binary, similar to (Ungless et al., 2023). Without this input, it is unclear how to identify relevant images or analyze stereotypes within them. This is also supported by our findings in Appendix E.2. Currently, automatic methods do not label images with "nonbinary", and as mentioned above, we are not aware of any other techniques that enable automatic analysis of images that may depict non-binary gender identities.

***Automatic gender labeling***. Using automatic methods to assign sensitive attributes such as gender (as well as race or age) can be problematic because models may introduce errors, carry their own biases, and in doing so, undermine the validity of analyses based on automatic labels. Even worse, if models are biased, they may reinforce those biases throughout the analysis. At the same time, using automatic tools is essential for conducting large-scale studies like ours. Therefore, we take steps to ensure our results are as valid as possible by addressing issues that arise from automatic methods. First, we filter images based on detected people, using state-of-the-art object detectors (Wang et al., 2024). Then, we crop person bounding boxes to reduce bias from background or contextual elements. Most importantly, we evaluate whether gender assignments from `InternVL2-8B` align with human annotations of perceived gender. As shown in Appendix E.1, this is indeed the case. Given the near-perfect alignment between human labels and automatically determined labels, we do

not expect automatic methods to introduce significantly more errors or reinforce stereotypes beyond what human annotators might. While gender bias remains a concern in MLLMs (Girrbach et al., 2025), it is less pronounced in discriminative tasks that aim specifically to label gender.

***Debiasing methods.*** The aim of our study is to provide a detailed, in-depth analysis of gender bias in current T2I models across everyday scenarios. In addition to understanding the societal issues related to T2I models, exploring ways to address these problems is also an important area of research. However, as models continue to be used without explicit steering mechanisms (Zhang et al., 2023; De Simone et al., 2023; Clemmer et al., 2024; Brack et al., 2023), it becomes crucial to develop a clear understanding of their underlying issues. Determining how and when to apply steering or other debiasing techniques is another complex challenge, which lies beyond the scope of this study. For instance, it remains an open question whether solutions to these identified problems should be implemented by model providers or users. One possible approach is "ambiguity in, diversity out" (Katirai et al., 2024), although this too raises concerns, such as maintaining contextual appropriateness. Given these challenges, detailed insights into model biases, like those provided in our study, are essential for making informed decisions about modifying or restricting model outputs. For the same reason, we do not aim to develop a benchmark. The fact that models exhibit bias has been shown before, and benchmarks typically construct one or a few measures of bias that help guide researchers and developers toward creating less biased models. However, such benchmarks can only indicate the degree of bias, not the specific manifestations of bias that we provide in this study.

Nonetheless, our findings are important and will inform future development of debiasing strategies. The main areas where our findings are relevant are *Targeted Data Curation*, as our fine-grained analysis identifies specific concepts and contexts (e.g., "laundry", occupations involving physical work) where gender bias is most severe. This can inform mitigation efforts like targeted data collection or strategic rebalancing of existing training sets. Also, our work informs *Hybrid Mitigation Strategies* targeting data and models alike. Here, our finding that models can amplify bias present in web-scale data (LAION-400M) is a crucial insight, because it implies that data-centric approaches alone may be insufficient. Future work should investigate hybrid strategies that address bias in both the training data and the training/inference process of the model.

***Social categories beyond gender.*** We find that T2I models show strong biases in other social categories, such as race and age, when generating images from the underspecified prompts used in our study. To illustrate this, we detect perceived race and age for all identified people in the generated images using `InternVL2-8B`. The prompts used are similar to those employed for detecting perceived gender. For detecting race and age, we use the following prompts:

```
What is the race of the person in the image?
A. black
B. east asian
C. indian
D. middle eastern
E. latino-hispanic
F. southeast asian
G. white
H. other
I. unclear/cannot tell
Answer with a letter (A, B, C, etc.).
```

And for age, we use the following prompt:

```
What is the age of the person in the image?
A. less than 20
B. 20-29
C. 30-39
D. 40-49
E. 50-59
F. 60-69
G. more than 70
H. unclear/cannot tell
Answer with a letter (A, B, C, etc.).
```

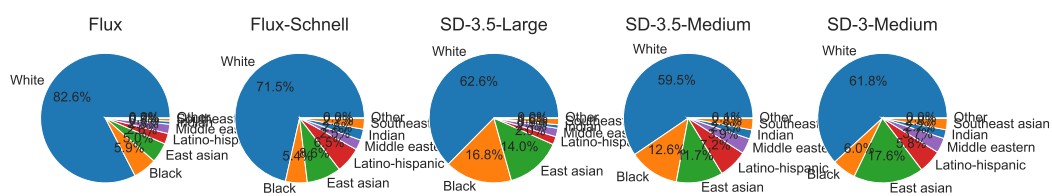

Figure 18: Race ratios for people in all images generated by T2I models, as assigned by `InternVL2-8B`.

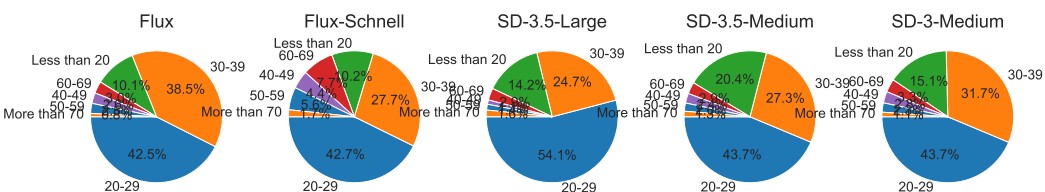

Figure 19: Age ratios for people in all images generated by T2I models, as assigned by `InternVL2-8B`.

In both cases, we randomly permute the option order (but not the option letters) to avoid label bias. Race and age categories are from FairFace (Karkkainen & Joo, 2021). However, we truncate the underage age categories to a single label ("less than 20").

We then calculate the overall ratios of people assigned to each race and age category for all 5 models in this study. Before calculating ratios, we drop all people who receive the "unclear/cannot tell" label. Results for race are in Fig. 18 and for age in Fig. 19. From these results, it is clear that models predominantly generate white and young (age 20-29 or 30-39) people, confirming results in (Wu et al., 2024; Ghosh & Caliskan, 2023).

In Table 10, we also show the top 10 prompts with the highest average ratio of generated people across models for each race. There, we find that White and East Asian individuals have a notable number of prompts that, consistently across models, generate predominantly images of the respective race in all T2I models. Moreover, only prompts associated with White people tend to be fairly general, while prompts linked to other races are mostly tied to cultural or national stereotypes. For example, East Asian-looking people are generated from prompts mentioning East Asian cultural elements, such as "anime" or "mandarin chinese", while Latino-looking people appear in images generated from prompts like "tacos" or "burrito". An analysis of such cultural stereotypes in T2I models has been conducted in (Dehdashtian et al., 2025; Jha et al., 2024).

Based on these findings, we conclude that the dominance of young, White individuals in generated images makes it difficult to perform intersectional analysis under the current experimental settings. To properly study race and age biases, as well as their intersection, it is necessary to explicitly prompt T2I models for these attributes and analyze the resulting stereotypes.

Lastly, we validate the performance of MLLM race and age detection using human annotations from the FairFace dataset. Using the prompts described above, we assign race and age labels to all images in the FairFace validation set. Detailed results are shown in Table 11. Overall, the accuracy is 68% for race and 56% for age. While these values are lower than the reported accuracies for gender, they are still significantly better than random chance, considering the larger number of categories. Therefore, we conclude that our observations about race and age stereotypes are approximately accurate, although a fine-grained analysis remains difficult due to the lower agreement between automatic methods and human labels.

## K    DETAILED COMPARISON TO PREVIOUS WORK

In this section, we provide a detailed comparison between our work and previous studies on analyzing gender bias in T2I models. We focus on works that use gender-neutral prompts, as this matches

|  | Precision | Recall | F1-Score | Support |
|---|---|---|---|---|
| Black | 0.83 | 0.91 | 0.87 | 1556 |
| East Asian | 0.67 | 0.86 | 0.75 | 1550 |
| Indian | 0.83 | 0.65 | 0.73 | 1516 |
| Latino-Hispanic | 0.56 | 0.53 | 0.55 | 1623 |
| Middle Eastern | 0.62 | 0.54 | 0.57 | 1209 |
| Southeast Asian | 0.58 | 0.48 | 0.53 | 1415 |
| White | 0.75 | 0.74 | 0.74 | 2085 |

(a) Detailed race labeling results by `InternVL2-8B` wrt. human annotations on the FairFace validation set. Overall accuracy is 68%.

|  | Precision | Recall | F1-Score | Support |
|---|---|---|---|---|
| 20-29 | 0.66 | 0.60 | 0.63 | 3300 |
| 30-39 | 0.48 | 0.45 | 0.47 | 2330 |
| 40-49 | 0.48 | 0.22 | 0.30 | 1353 |
| 50-59 | 0.42 | 0.37 | 0.39 | 796 |
| 60-69 | 0.30 | 0.56 | 0.39 | 321 |
| less than 20 | 0.81 | 0.82 | 0.81 | 2736 |
| more than 70 | 0.34 | 0.60 | 0.43 | 118 |

(b) Detailed age labeling results by `InternVL2-8B` wrt. human annotations on the FairFace validation set. Overall ccuracy is 56%.

Table 11: Race and age classification results on the FairFace validation set.

the experimental setup in our study. The comparison is shown in Table 12. For each paper, we include the number of gender-neutral prompts, the total number of images generated per evaluated model, a brief summary of the main findings, and a short note on how our study differs from that work.

In comparison to previous work, our study significantly improves the understanding of gender bias in T2I models by offering a detailed analysis across a wide range of everyday activities, places, objects, and occupations. As noted by Wan et al. (2024) and clearly shown in Table 12, most prior studies have focused mainly on occupational prompts to highlight bias. While this focus is valuable, examining gender bias beyond occupations is also essential for a more complete understanding of how such bias manifests in T2I models.

Another aspect is the typically very small scale of studies, as also shown in Table 12. While this allows us to conclude that models are biased, gaining concrete insights into these biases requires a broader analysis like ours. Two other studies also generate a large number of images: Wu et al. (2024) generated images from 200,000 distinct prompts, but used them not to analyze gender distributions for individual prompts or prompt groups, but to examine representational similarities between images from gender-neutral and gendered prompts. This setup is well-suited to reveal an overall male bias in the evaluated models, but does not support a detailed analysis of the specific stereotypes replicated by the models. Similarly, Luo et al. (2024; 2025) (these two papers have significant textual overlap) generated images for 2,123,200 prompts, about 70% of which focus on occupations. This study uses the images to compute holistic bias scores for comparing and ranking models, whereas our goal is to document biases in detail.

| | # Images | # Prompts | Main Findings | Novelty of our work |
|---|---|---|---|---|
| Bianchi et al. (2023) | 2000 | 20 | This study includes 20 gender-neutral prompts (in addition to prompts that either explicitly specify gender and race or focus on objects from diverse cultural contexts). Of these, 10 prompts describe people (e.g., "an exotic person", "a terrorist"), and 10 describe occupations. The paper reports on how gender and racial stereotypes are reflected and perpetuated in these 20 analyzed cases. | Our study enables a more thorough analysis of gender bias in T2I image models by including a larger set of prompts and generated images. This makes it possible to automatically evaluate broader trends, such as associations with household chores or workplaces, beyond just a few manually examined examples. |
| Cheong et al. (2024) | 1050 | 105 | Generated 10 images each for 105 different occupations. After collecting gender and race annotations from human labelers, a filtered set of 67 occupations was compared with respect to race and gender ratios in the U.S. workforce and the generated images. The study finds strong bias amplification, i.e., the images often depict only men or only women for a given occupation. | Our study examines gender bias not only in occupations but also in related categories such as activities, places, and objects. Within occupations, we include the complete set from the U.S. BLS list. Our method produces more reliable estimates of gender ratios by sampling a larger number of images and filtering out unsuitable prompts and images. |
| Chinchure et al. (2024) | 5328 | 111 | This study proposes a method to detect biases in generated images on dynamically identified bias axes. First, an LLM proposes potentially relevant bias axes from the given prompt. Then, after a set of images has been generated from the prompt, these biases are verified or rejected based on counterfactual image sets and VQA attribute detections. The relevance of the detected biases is verified through human judgments. | In contrast to proposing a general method to discover various biases, we specifically analyze gender bias regarding a broad range of activities, contexts, objects, and occupations. We see our research and the method proposed in Chinchure et al. (2024) as orthogonal and mutually beneficial: Once systematic biases in T2I models have been detected through methods like TIBET, large-scale analyses such as those conducted in our study will give a deep understanding of how exactly these detected biases surface. |

| Cho et al. (2023) | 747 | 83 | Generated 9 images based on 83 gender-neutral occupation prompts (excluding variants that explicitly specify gender). Gender, skin tone, and 15 other attributes were automatically detected. The results show that T2I models generally generate more men than women. Additionally, skirts appear only on women, while suits are more commonly shown on men. | Our study analyzes gender bias not only in occupation-related prompts but also in everyday activities and locations. In addition, our evaluation protocol provides a more reliable estimate of gender ratios by sampling more images and filtering out unsuitable ones. Lastly, we reduce contextual bias in automatic gender detection by cropping the images to focus on person bounding boxes. |
|---|---|---|---|---|
| Luccioni et al. (2024) | 4380 | 146 | Generated 30 images using 146 gender-neutral occupation prompts. Gender and race distributions were analyzed with a non-parametric method that does not rely on explicit gender or race labels. A comparison with U.S. BLS statistics shows that women — especially Black women — are underrepresented. | We analyze gender biases beyond just occupations while also including a larger set of occupations. This broader analysis helps us identify bias trends on a wider scale. At the same time, we ensure our results are reliable by using large-scale sampling and filtering. |
| Luo et al. (2024; 2025) | 2 123 200 | 2654 | Generated 2,654 prompts related to occupations, social relationships, and attributes. A significant portion of the prompts include explicit gender or race identifiers, and about 70% of the prompts focus on occupations. The study evaluates different models using various bias scores to determine which ones are the most or least biased. | While this study also provides a large-scale evaluation of bias in T2I models, our work offers two key contributions: First, instead of presenting overall bias scores to compare models, we closely examine which specific biases (e.g., those related to household chores) the models exhibit. Second, our study goes beyond occupations, which are the main focus of the FaintBench benchmark, and explores a broader range of categories. |
| Lyu et al. (2025) | 2000 | 100 | This study evaluates the reliability and validity of gender bias analysis pipelines, identifying several issues, such as images featuring people of different genders or no people at all. The prompts used in the analysis cover all the categories included in this study, but on a much smaller scale (10 to 40 prompts). | Our study focuses on a detailed analysis of gender bias in T2I models at a large scale. To achieve this, we include a significantly higher number of prompts and analyses, comparing our results to those related to human stereotypes. However, the insights from this study shaped our experimental design, particularly emphasizing the need for careful and rigorous filtering. |

| | | | | |
|---|---|---|---|---|
| Seshadri et al. (2024) | 31000 | 62 | This study investigates bias amplification using 62 occupation prompts and concludes that bias amplification is largely explained by distribution shifts between the training and probing distributions. | Our study thoroughly documents the gender bias in recent T2I models, including observations of bias amplification. However, we do not explore the causes behind this bias amplification. Instead, we analyze a broad range of activities, places, objects, and occupations to provide in-depth insights. |
| Ungless et al. (2023) | 924 | 231 | Generated 4 images for each of 321 prompts centered on non-binary identities. The key findings are that non-binary identities are poorly represented by T2I models, often resulting in the creation of NSFW or degrading content. | Our study focuses specifically on binary gender. We also note that, without explicit instructions, models do not produce images that clearly represent non-binary identities. As a result, it is currently impossible to quantitatively explore biases related to non-binary identities using the models and methods applied in this study. However, we believe that addressing this issue is an important direction for future research. |
| Wu et al. (2024) | 800 000 | 200 000 | This study examines how gender-neutral prompts are represented across different T2I models (text, latent noise, and images). The key finding is that when gender (or other image characteristics) is not specified in the prompt, the generated images tend to resemble those created from masculine prompts. | While this study analyzes a large number of prompts, it does not estimate gender ratios for prompts or prompt groups. Instead, it focuses on examining representational similarities. In contrast, our method allows for a deeper exploration of biases across a wide range of activities, places, objects, and occupations. This approach enables us to make precise statements about whether models display specific types of gender bias. |

Table 12: Detailed comparison to previous work. We show the number of images in the study for each evaluated model, the number of prompts, a summary of the study's findings, and a comment on how our study contributes beyond the respective prior work.

