# OpenReview forum: "A Large Scale Analysis of Gender Biases in Text-to-Image Generative Models"
_ICLR.cc/2026/Conference — ICLR 2026 Conference Withdrawn Submission_

### Official Review · Reviewer_DbY9 · 2025-10-28

**Soundness:** 2
**Presentation:** 2
**Contribution:** 1
**Rating:** 2
**Confidence:** 4

**Summary:**

This paper presents a large-scale computational analysis of gender bias in five recent text-to-image (T2I) models. The authors move beyond the commonly studied domain of occupations to also analyze biases in everyday activities, contexts, and objects. They create a new dataset of 3,217 gender-neutral prompts and generate over 3.2 million images, filtering them down to ~2.3 million for analysis. Using an automated pipeline to detect people and identify perceived gender, the study finds that T2I models consistently reinforce traditional gender stereotypes. For example, women are predominantly associated with care, household, and human-centered scenarios, while men are associated with technical or physical labor. The authors also provide evidence that models may amplify existing biases.

**Strengths:**

1. The study's primary strength is its large-scale analysis, examining over 2.2 million images from 3,217 prompts across five different models. This provides a statistically robust and comprehensive snapshot of the bias present in current SOTA open-source models.

2. The creation of a new dataset of gender-neutral prompts is a useful contribution. Expanding the analysis beyond occupations to include everyday activities, contexts, and objects provides a more holistic picture of how gender stereotypes are embedded in these models.

**Weaknesses:**

1. The major weakness is its limited originality. The finding that T2I models exhibit and amplify gender bias is not new; it is a well-established problem, which the paper acknowledges. The paper's main contribution is scaling up the evaluation size, which confirms what is already known rather than providing a new, fundamental insight.

2. This paper is purely an "analysis" paper. It meticulously documents a known problem but offers no mitigation method or algorithmic solution, stating this is "beyond the scope of this study". This limits its constructive impact for the research community.

3. The selection of models, while including important open-source baselines, is missing several current state-of-the-art closed and open models (e.g., gpt-image, BAGEL, nano banana). A "large-scale" study claiming to analyze the current landscape of T2I models would be more comprehensive if it included these highly-performant or widely-used models.

4. The paper's statement in Section 2 that "Luccioni et al. (2024) measure bias without explicit gender/race labels" (restated in Appendix K) appears incorrect. The cited paper (arXiv:2303.11408) explicitly states its analysis framework spans "identity characteristics — ethnicity and gender," which contradicts this claim.

5. The entire analysis pipeline hinges on an MLLM (InternVL) to perform gender classification. The human validation study for this classifier is insufficient, as it relied on only two human annotators (with no background information provided) and was evaluated on a very small sample of only 120 bounding boxes. This is not a robust enough validation for the central measurement tool of a 2.2-million-image study.

6. The paper fails to define a clear standard for what constitutes "gender" for its annotators (both human and VLM). It is unclear if the evaluation is based purely on visual appearance, stereotypical presentation, or other cues. This ambiguity is a significant methodological flaw.

7. The paper and appendix do not include any examples of the generated images. This makes it difficult for reviewers to qualitatively assess the model outputs and verify the claims. Including a sample of evaluated images is standard practice and essential for transparency.

8. The study operates on a strictly binary definition of gender (men and women) and explicitly excludes non-binary identities. While noted as a limitation, this omits a crucial axis of bias and harm.

**Questions:**

1. Given that gender bias in T2I models is a well-established finding, what is the single most important novel insight (not just a measurement) that your large-scale analysis provides which smaller-scale studies missed?

2. Can you justify the selection of the five models evaluated? Why were other highly-performant SOTA models (like gpt-image, BAGEL, etc.) omitted from this "large-scale" analysis?

3. Can you please clarify the standard for "gender" evaluation used? What instructions were given to the two human annotators and what criteria (e.g., purely appearance-based?) does the VLM use to determine perceived gender?

4. Can you clarify the statement regarding Luccioni et al. (2024)? Your paper claims they measure bias "without explicit gender or race labels", but their paper (arXiv:2303.11408) appears to use these explicit labels for its analysis. Please explain this discrepancy.

5. How can you robustly claim bias "amplification" when, as you note, "the overall ratio of men and women in the pretraining data remains unknown"? How do you disentangle amplification from the simple reflection of a (potentially) highly skewed training set?

---

### Official Review · Reviewer_F5U7 · 2025-10-30

**Soundness:** 3
**Presentation:** 3
**Contribution:** 2
**Rating:** 2
**Confidence:** 5

**Summary:**

The paper studies several image generation models for gender bias in four experimental setups for images depicting people while conducting activities, people in a given context, people in relationship with objects, and occupational representations. The study consists of a large number of prompts and generated images and can be considered a benchmark for studying such biases.

**Strengths:**

The paper studies an important topic and the results are clearly described.

**Weaknesses:**

- The findings of the study are not novel and studied in depth in other papers. I carefully read the comparison with related work in page 38 and had a very hard time to convince myself that there is a sufficient level of novelty in the method or findings. The study is perhaps more thorough in certain areas but by now there have been several works that reach similar conclusions. Another earlier study that also compares with BLS data is "Social Biases through the Text-to-Image Generation Lens", which also draws from previous work that uses BLS data as reference for studying biases of Image Search "An Image of Society: Gender and Racial Representation and Impact in
Image Search Results for Occupations.".

- What is somewhat more novel is the parallel analysis on LAION which the authors could consider emphasizing more. Another angle would be to consider image generation for mitigating present biases and challenges around such processes as they do not usually work out of the box.

**Questions:**

NA

---

### Official Review · Reviewer_4qUz · 2025-10-31

**Soundness:** 2
**Presentation:** 3
**Contribution:** 1
**Rating:** 2
**Confidence:** 4

**Summary:**

The paper investigates gender bias in T2I models, focusing on everyday scenarios rather than just occupations. They created a dataset of over 3k gender-neutral prompts and generated 2M images using five T2I models. They used automated methods to detect the perceived gender of people in the images, filtering out those without people or with mixed-gender groups, resulting in a dataset of over 2 million images. Prompts were grouped by similar concepts to analyze gender representation across different contexts. Findings show that T2I models tend to reinforce traditional gender roles and stereotypes.

**Strengths:**

- S1: The paper tackles and important problem which is ongoing although the issues have been present for quite some time already
- S2: Previous works mostly rely on standard classification tools and this work uses VLM-based approaches now which seems more timely. though it would be good to know what the difference between them is in terms of accuracy/biases.
- S3: The authors invest a lot of resources/efforts to investigate this issue.

**Weaknesses:**

- W1: The biggest weakness of the paper is its novelty and contribution to the field. Basically all findings are already known. E.g. Bianchi et al. or Friedrich et al. show exactly the same findings. Just arguing with increased scale, better filtering, and more prompts is not really convincing. The even more pressing question is, how does this work inform the field? What should the field do differently, how can we improve upon? What findings are specifically novel so the field can learn from it. Or is it the way we should evaluate our T2I models from now on? Clarifying your contribution more would be helpful.
- W1.1: The paper is also methodologically not novel. For example, the dataset to generation comparisons is also present in Seshadri et al and Friedrich et al. Furthermore, LAION-400M is not even the exact training data of the models.
- W2: all images are generated roughly from the same initial data distribution, i.e. stable diffusion and flux are very closely related models from similar authors. Their similarity is hence not so surprising. There are much more SOTA models out like Qwen-Image, also omni models like BAGEL, would be relevant instead of using 5 times similar models.
- W3: While understandable for consistency and validity, using such artificial prompts does not really reflect the model and the way they are used. T2I models are no lookup table and showing they are biased for those types of prompts does not necessarily carry over to other queries, e.g. also gender-specific prompts (reference: Friedrich et al. "Multilingual text-to-image generation magnifies ...").
- W4: The statistical significance of most experiments and hence their conclusions drawn remains limited.

**Questions:**

- Why exactly did you filter images that have multiple people of multiple genders?

**Details Of Ethics Concerns:**

ethical concerns have been addressed and discussed as necessary

---

### Official Review · Reviewer_8dTW · 2025-11-01

**Soundness:** 3
**Presentation:** 2
**Contribution:** 2
**Rating:** 2
**Confidence:** 4

**Summary:**

* The study compiles a dataset of 3,217 gender-neutral prompts across four categories (Activities, Contexts, Objects, and Occupations) to conduct a large-scale analysis of gender bias. It extends bias analysis beyond the typical focus on occupations to include everyday activities, contexts, and objects.
* The authors generated 200 images for each prompt (with 5 variations) from 5 leading open-source T2I models, resulting in a dataset of over 2.29 million filtered images for analysis.
* The analysis systematically shows that T2I models reinforce traditional gender roles and stereotypes. Specifically, women are predominantly portrayed in care and human-centered scenarios, while men are shown in technical or physical labor contexts.

**Strengths:**

* The proposed dataset includes analysis of bias in novel settings like daily activities.
* This large scale experiments enables more precise insights and reduces limitations seen in smaller-scale studies.
* The proposed dataset reveals gender bias that stereotype women in traditional roles like homemakers and in caring activities, while men are stereotyped to do physical work or machinery.

**Weaknesses:**

While the proposed benchmark and experiments are mostly solid, my main concern is on the novelty and significance of contribution of the paper. Please see bullet points below:

* **Limited Novelty and Perspective**: While the scale of the analysis is significant, the central finding—that TTI models exhibit and amplify known gender stereotypes (especially in occupations and traditional household roles)—is not novel. The work primarily validates existing conclusions but on a larger dataset. Other research has explored similar, even more complex dimensions of TTI bias and proposed alternative benchmarks. The authors need to justify how their work is more novel, how their contribution is different from these works, and how their contribution is significant.
  * [1] examined how different genders are depicted (presentational bias), such as in posture, facial expression, and clothing.
  * [2] explores gender bias in novel aspects such as stereotypical correlation with certain subjects (e.g. astronomy)
  * [3] explores gender bias in novel 2-subject generation scenarios, including exploration on a novel bias dimension (organizational power).
* **Insufficient Model Evaluation**: The study's conclusion regarding the state of bias in TTI models is weakened by its reliance on older and only open-source models (e.g., Flux, Stable Diffusion 3.5 Large/Medium, SD-3 Medium). The paper omits evaluation of leading state-of-the-art models, particularly powerful closed-source systems like DALL-E 3 and GPT-4 Image, and recent high-performing open-source models like Qwen-Image. Their inclusion is essential for providing a current, comprehensive assessment of the field.
* **Lack of Verification for Automatic Gender Labeling Pipeline**: The core evaluation process, which relies on a multi-step pipeline involving automated bounding box detection and the InternVL system for automatic gender identification, lacks necessary empirical verification. The authors need to provide at least one of the following to justify the validity of evaluation pipeline:
  * Quantitative results on a recognized demographic recognition benchmark.
  * A human validation study (e.g., a crowd-sourced audit) to confirm the accuracy of the MLLM's labels on a subset of the generated images, which is standard practice in bias research to ground automatic metrics in human perception.

[1] Sun, Luhang, et al. "Smiling women pitching down: auditing representational and presentational gender biases in image-generative AI." Journal of Computer-Mediated Communication 29.1 (2024): zmad045.

[2] Wang, Jialu, et al. "T2IAT: Measuring Valence and Stereotypical Biases in Text-to-Image Generation." Findings of the Association for Computational Linguistics: ACL 2023. 2023.

[3] Yixin Wan and Kai-Wei Chang. The Male CEO and the Female Assistant: Evaluation and Mitigation of Gender Biases in Text-To-Image Generation of Dual Subjects. ACL 2025. 2025.

**Questions:**

Please see Weakness.

---

### Official Review · Reviewer_9tQG · 2025-11-03

**Soundness:** 2
**Presentation:** 3
**Contribution:** 2
**Rating:** 2
**Confidence:** 5

**Summary:**

This paper conducts a large-scale analysis of gender bias in text-to-image generative models. The authors generate a dataset comprising 2.3 million images from 3,217 gender-neutral prompts using five state-of-the-art open-source models (Flux, Flux-Schnell, and Stable Diffusion 3/3.5). Their analysis spans four domains: Activities, Contexts, Objects, and Occupations.

**Strengths:**

- The authors present a large-scale dataset comprising 2.3 million images from 3,217 gender-neutral prompts using five state-of-the-art open-source models (Flux, Flux-Schnell, and Stable Diffusion 3/3.5).
- The analysis covers a wider range of domains and environments not examined in prior studies, including activities (e.g., cooking, sports), contexts (e.g., school, factory), objects (e.g., jewelry, vehicles), and occupations (derived from the U.S. Bureau of Labor Statistics).

**Weaknesses:**

- A key limitation of this work lies in its narrow scope. While addressing gender bias is valuable, it represents only one dimension of bias in text-to-image models. Strong performance on a gender bias benchmark does not imply that a model is free from other types of bias (e.g., racial, cultural, or age-related). Thus, while the study offers useful insights for researchers specifically examining gender bias, its relevance to a broader audience concerned with overall model fairness may be limited.

- The study lacks an intersectional and causal perspective, for instance, see [1]. Although race, age, and cultural factors are briefly mentioned in the appendix, they are not systematically analyzed alongside gender. Moreover, while the paper effectively documents bias patterns, it does not investigate the underlying mechanisms, such as data imbalance, prompt encoding, or model architecture, which may give rise to these biases.

[1] Mitigate One, Skew Another? Tackling Intersectional Biases in Text-to-Image Models

**Questions:**

- Did you consider whether similar bias amplification effects might appear in non-gender dimensions?

- How might your methodology be extended to support intersectional analysis (e.g., gender × race interactions)?

- How reliable is the gender classification from InternVL2-8B across diverse demographic attributes (e.g., race, lighting, attire)? Have you validated its predictions with human annotators?

- Did you observe any relationship between model size and bias consistency across domains (beyond the slight reduction in SD-3.5-Large)?

---

### Official Review · Reviewer_2KS7 · 2025-11-03

**Soundness:** 3
**Presentation:** 3
**Contribution:** 2
**Rating:** 6
**Confidence:** 4

**Summary:**

This paper presents an audit of gender bias in T2I models, extending prior occupation-focused studies to a broader set of real-world situations. The authors analyze 1,405 activities, 737 contexts, 500 objects, and 575 occupations (identified in previous work) generating a total of 3.2 million images (200 per prompt) using five open-source T2I models.

The gender is identified using two successive models, YOLOv10 for cropping regions for person detection, followed by InternVL2-8B for binary gender classification on the cropped regions.

The analysis reveals several key findings:
	1.	Models default to depicting men, even for gender-neutral prompts.
	2.	Clear stereotype patterns emerge — activities such as care work, shopping, yoga, and baking for females, while outdoor, technical, transport, and gaming prompts for males.
	3.	Generally, amplifications of gender bias extends beyond occupations to everyday activities: compared to images from LAION-400M, the percentage of depicted females drops from 52% to 41% in generated images, with males increased to 65–87%.

**Strengths:**

- The paper is well-motivated, clearly structured, and easy to follow.
- It presents a systematic taxonomy of prompts, grounded in prior work ([1]), to analyze different real-world situations.
- The experimental design and analysis are thorough, including comparisons of gender distributions against real-labor statistics (even though these rely on prior datasets).
- The prompt sets are carefully designed, clustered, and analyzed — the presentation (e.g., Figure 3) effectively conveys the key findings.
- The study employs multiple text-to-image (T2I) models, enhancing generality and cross-model validity.
- The semantic clustering approach (building on [1]) is well-discussed in light of previous literature, allowing fine-grained interpretation of sub-clusters.
- The findings consistently reveal strong stereotype alignment across models, indicating robustness of the observed biases.

[1] Steven Wilson and Rada Mihalcea. Measuring semantic relations between human activities. In IJCNLP, 2017.

**Weaknesses:**

- The activity taxonomy is based solely on U.S. data, which limits cultural generalizability.
- Excluding mixed-gender or low-frequency (<100/200) prompts may artificially “clean” the dataset and bias cluster representation (Images containing no person, unclear gender, or mixed genders are excluded, resulting in 2.29 million valid samples.).
- The analysis remains coarse-grained, with limited coverage of attributes such as skin tone, age, region, and non-binary identities.
- The paper could use ablation studies (e.g., testing with a second classifier, using counter-stereotypical prompts, or assessing model behavior under prompt perturbations) to make the findings more robust.
- The resulting dataset, while size increased to previous work, is still limited in size, as it is derived from a small number of base prompts to generate to 200 images per model.
- No human evaluation is reported to assess the semantic quality or retrieval fidelity of the clusters.

**Questions:**

Could you test come counter-stereotype prompts, if not already done, e.g., for car repair, baby care, test whether explicit female/male tokens are able to flip the default across models?
Did you test another classification method, e.g., CLIP-based, or closed-source (GPT)?

---

### Note · Authors · 2025-11-14

**Comment:**

We sincerely thank the reviewers for their time and valuable comments.

**Withdrawal Confirmation:**

I have read and agree with the venue's withdrawal policy on behalf of myself and my co-authors.